# Estimates of tree root water uptake from soil moisture profile dynamics

Conrad Jackisch[1,2], Samuel Knoblauch[1,3], Theresa Blume[4], Erwin Zehe[1], and Sibylle K. Hassler[1]

[1]Karlsruhe Institute of Technology (KIT), Institute of Water and River Basin Management, Chair of Hydrology, Kaiserstr. 12, 76131 Karlsruhe, Germany.
[2]Technische Universität Braunschweig, Institute of Geoecology, Dept. Landscape Ecology and Environmental Systems Analysis, Langer Kamp 19c, 38106 Braunschweig, Germany.
[3]University of Greifswald, Dept. of Biology, Friedrich-Ludwig-Jahn-Str. 15, 17489 Greifswald, Germany.
[4]Helmholtz Centre Potsdam, GFZ German Research Centre for Geosciences, Section Hydrology, Potsdam, Germany.

**Correspondence:** Conrad Jackisch (c.jackisch@tu-braunschweig.de)

**Abstract.** Root water uptake (RWU) as an important process in the terrestrial water cycle can help to better understand the interactions in the soil-plant-atmosphere continuum. We conducted a field study monitoring soil moisture profiles in the rhizosphere of beech trees at two sites with different soil conditions. We present an algorithm to infer RWU from step-shaped, diurnal changes in soil moisture.

While this approach is a feasible, easily implemented method for moderately moist and homogeneously textured soil conditions, limitations were identified during drier states and for more heterogeneous soil settings. A comparison with time series of xylem sap velocity underlines that RWU and sap flow (SF) are complementary measures of the transpiration process. The high correlation between the SF time series of the two sites, but lower correlation between the RWU time series, suggests that soil characteristics affect RWU of the trees but not SF.

*Copyright statement.* TEXT

## 1   Introduction

Evapotranspiration (ET) is a key water and energy flux in ecosystems. Although ET amounts globally to 60% of total precipitation in terrestrial systems (Oki and Kanae, 2006) and transpiration is claimed to dominate the terrestrial water cycle (Jasechko et al., 2013), it remains one of the most challenging fluxes to observe and understand (Wulfmeyer et al., 2018; Renner et al.,
2019). ET describes the release of water vapour into the atmosphere, driven by the saturation deficit of the atmosphere and influenced by soil and vegetation characteristics, which control soil water uptake and transport. It can be either limited by the radiative energy supply or by the terrestrial water supply.

Evaporation is studied using experiments and models (e.g. Shuttleworth, 2007; Or et al., 2013). Transpiration is a more complex interplay of different fluxes including root water uptake (RWU) and sap flow (SF). It is well known that the controls
of transpiration are not static (Renner et al., 2016; Dubbert and Werner, 2019). Plants can adapt their water uptake and transport

to their assimilation under different stressors (Schymanski et al., 2009; Lu et al., 2020). Additionally, plants can store water to buffer intermediate stresses (Cermak et al., 2007; Gao et al., 2014), resulting in deviations between RWU and SF. Studies on plant transpiration frequently focus on stomatal control (Schymanski and Or, 2017) and theories on leaf-related dynamics and the transpiration loss function (Sperry and Love, 2015). To estimate transpiration of individual trees, SF measurements are widely used (e.g. Nadezhdina et al., 2010; Poyatos et al., 2016). However, a series of approximations and assumptions is needed to convert the sap velocity to the volumetric water flux in a tree or stand.

RWU is a missing link to understand water limitation, as it taps the soil water store and is most difficult to observe. Accordingly, comparably few studies and measurement standards exist. For small plants, lysimeters are one means to quantify how plants control ET (e.g. Gebler et al., 2015). Moreover, details about the shape of the rhizosphere can be revealed with tomographic analyses (e.g. Kuhlmann et al., 2012; Pohlmeier et al., 2017), but not necessarily about the dynamic RWU process in the rhizosphere. At larger scale and for larger plants, changes of groundwater levels (e.g. Maxwell and Condon, 2016; Blume et al., 2018), isotope signatures of water (e.g. Dubbert and Werner, 2019) and carbon (e.g. Vidal et al., 2018), and SF measurements in the roots have been employed (e.g. Burgess et al., 2000). To understand RWU, a series of approaches to measure (e.g. Mary et al., 2016) and simulate (e.g. Pagès et al., 2004; Javaux et al., 2008) the root architecture and its interaction with soil hydrology have been developed. Among these are representations based on resistance terms (e.g. Couvreur et al., 2012) or based on thermodynamic optimality through a minimisation of physical work during root water uptake (Hildebrandt et al., 2016). It is known that RWU responds to soil water conditions (Cai et al., 2018) and thus soil structure. Additionally, studies found that roots and soil structure co-evolve (Carminati et al., 2012) and that roots can actively modify the soil properties by mucilage (Carminati et al., 2016; Kroener et al., 2018).

So far, only few examples for quantitative in-situ observations of tree RWU dynamics exist (e.g. Rodríguez-Robles et al., 2017; Leuschner et al., 2004). Approaches based on an analysis of stable isotopes in the rhizosphere and the plant xylem can identify the path of the water from different soil depths into different parts of the tree in great detail (Dubbert and Werner, 2019; Zarebanadkouki et al., 2019). From a soil perspective, the complex effect of RWU can be observed as a decrease of soil water content during active water transport through plants (Novák, 1987; Feddes and van Dam, 2005; Guderle and Hildebrandt, 2015), but technologies for spatially distributed measurement of soil moisture dynamics at relevant scales are just emerging (Klenk et al., 2015; Allroggen et al., 2017; Jackisch et al., 2017; Boaga et al., 2013). However, there has not been much research on how well this diurnal decrease reflects the water transport into and within trees.

A change of soil moisture is not necessarily RWU, SF and eventually transpiration. It can also be caused by hydraulic redistribution within the soil (Burgess et al., 1998). Similarly, temporary water storage in the tree's hydraulic system can lead to SF and transpiration without the corresponding RWU (Cermak et al., 2007; Matheny et al., 2015). Hence, studying the spatio-temporal dynamics of soil-moisture-derived RWU and its correlation to SF might be key to more holistic observations (Jackisch et al., 2017; York et al., 2016) of forest water dynamics including the main actors (the trees, Ellison et al., 2017). In that sense, spatially distributed monitoring of both RWU from soil moisture and SF could help to elucidate differences between the influence of the geological and pedological settings on water supply and the influence of the plants themselves, i.e. their adaptations in root systems, dynamic sourcing of water (Nadezhdina et al., 2010) and transpiration regulation (Lu et al., 2020).

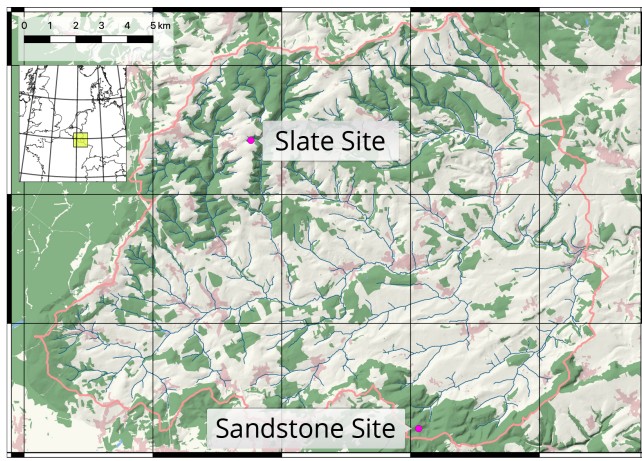

**Figure 1.** Attert experimental basin, Western Luxembourg. Locations of the two reference sites. Basemap: Landcover from OpenStreetMap contributors. Shading and river network calculated with a combined DEM of the administrations of Luxembourg and Wallonie.

The aim of this study is to evaluate the potential and limitations of estimating RWU from the diurnal decrease of rhizosphere soil moisture (Guderle and Hildebrandt, 2015; Guderle et al., 2018) in forest systems. We structure our analysis along the following research questions:

**1.** Can daily RWU be robustly derived from records of soil moisture dynamics?

**2.** Are the dynamics in derived RWU consistently related to dynamics in SF?

**3.** How do soil and site characteristics affect RWU and SF?

For this analysis we develop and assess an automated approach to derive RWU estimates from soil moisture profile measurements. We compare the RWU dynamics to SF measurements in two beech stands of different geological and pedological setting but with very similar weather, climate and topography. The developed calculation algorithm is published as python package *rootwater* under MIT license (Jackisch, 2019).

## 2  Field sites and monitoring

In the vegetation period of 2017, we selected and instrumented two sites in mixed beech stands (*Fagus sylvatica*) in contrasting geological settings, one on loamy sand in a sandstone basin (sand site) and another on loamy regosol on periglacial coverbeds of the slate Ardenne Massif (slate site, Fig. 1). Both sites are located in the Attert experimental watershed in western Luxembourg and are part of the monitoring setup within the CAOS research unit (Zehe et al., 2014). The climate is temperate semi-oceanic, mean annual rainfall is 845 mm (Pfister et al., 2014) and mean monthly temperatures range between 0°C in January and 17°C in July (Wrede et al., 2015).

## 2.1 Soil moisture monitoring

Soil moisture was monitored using a sequence of TDR tube probes (Pico Profile T3PN, Imko GmbH), which allow for installation with minimal disturbance using an acrylic glass access liner (diameter $48\,\mathrm{mm}$). The liner tube was installed in the rhizosphere of the trees without any excavation using a percussion drill (about $0.5\,\mathrm{m}$ from the stem). For optimal contact of the liner with the surrounding soil, the drill diameter was $40\,\mathrm{mm}$ and the tube was installed more than one year prior to the recorded data set. Each TDR probe segment integrates the soil moisture measurement over its length of $0.2\,\mathrm{m}$. The signal penetrates the soil about $0.05\,\mathrm{m}$ which results in an integral volume of approx. $1\,\mathrm{L}$. The probes are stacked directly on top of each other, permitting spatially continuous monitoring over the soil moisture profile.

At the sand site, we were able to install a profile with a sequence of 12 probes reaching a depth of $2.4\,\mathrm{m}$. At the slate site, percussion drilling was inhibited by the weathered bedrock. There we could only install a profile with a sequence of 9 probes reaching a depth of $1.8\,\mathrm{m}$.

## 2.2 Soil hydraulic characteristics of the sites

The sand site is located in the Huewelerbach subbasin which is characterised by deep, homogeneous sandy soils and deep groundwater-driven hydrology. The second site on regosol of the slate Ardenne Massif is located in the northern part of the Colpach subbasin (Fig. 1). It is characterised by high gravel content and inter-aggregate voids (Jackisch et al., 2017). In this area the hydrological regime is dominated by a flashy response to rainfall through macroporous soils (Glaser et al., 2019).

The two sites show contrasting hydrological characteristics. An exemplary event water balance, based on above-canopy precipitation and the change in soil moisture in the different depth layers, is given in Figure 2. While both sites show about 30% of the event water being stored in the soil after five days, the response of the soil profiles to the water input is very different between the sites.

At the sand site (Fig. 2A), the fraction of the precipitation which is not intercepted in the canopy and litter layer enters the top soil horizon and successively percolates through the soil profile. This can be seen as diagonal patterns. The overall event water balance remains roughly constant. These dynamics are coherent with an expected event reaction of an ideal porous medium. Here, we can reasonably assume to represent the rhizosphere soil water dynamics in our profile measurements.

At the slate site (Fig. 2B), the same event causes a fast response in deeper soil layers with an initial overshoot of the water balance and a quick recession. This suggests a non-uniform infiltration process, followed by diffusive lateral redistribution into the surrounding soil. The latter can be seen as simultaneous declines of soil moisture in the different depth layers. The hydrological regime at this site is dominated by flashy transport through the macroporous soils and fill-and-spill mechanisms of subsurface pools on the fissured bedrock (Jackisch, 2015; Loritz et al., 2017).

Since soil moisture is measured as dielectric permittivity of the bulk soil, the measurement principle integrates over the entire soil volume, irrespective of stone content, voids or wetted contact surfaces. The joints and fractures of the weathered bedrock at the slate site add two restrictions to representative soil moisture measurements: i) Roots are likely to grow along these fractures where event water will be stored with little effect on the bulk soil moisture. ii) Rocks inhibiting the drilling

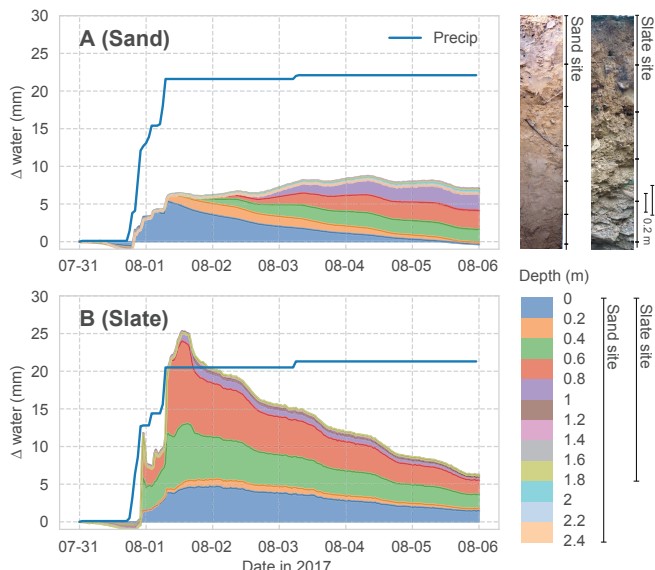

**Figure 2.** Event water balance observed at both sites. Shown is the stacked change of soil water content in each monitored soil depth. Cumulated above-canopy precipitation input is given as the blue line. The other colours correspond to the different depths in the soil profiles as shown in the figure legend. Two profile images give an impression of the soil conditions at the two sites.

prevented us from sampling the entire rooting depth. Hence, the soil moisture measurements are prone to miss parts of the active rhizosphere at this site.

### 2.3 Sap velocity and meteorological data

SF sensors were installed in several trees at breast height before leaf out of the vegetation period in 2017. At the sand site, the reference sap velocity time series for this study could be obtained from the beech tree closest to where the TDR sensors were installed. It had a diameter at breast height (DBH) of $64\,\mathrm{cm}$ and was approximately $0.5\,\mathrm{m}$ away from the TDR tube. At the slate site, the sap velocity sensor of the intended tree failed 3 weeks after leaf out. There, we refer to a neighbouring beech tree with a DBH of $48\,\mathrm{cm}$ about $9\,\mathrm{m}$ from the TDR measurements (see Appendix A for details). The SF sensors we used (East30 Sensors) are based on the heat ratio method and measure simultaneously at 5, 18 and 30 $\mathrm{mm}$ depth within the sapwood. Installation and calculation of sap velocities followed the description in Hassler et al. (2018).

As further reference for the drivers of temporal dynamics in soil moisture and sap velocity we use solar radiation records (Apogee Pyranometer SP110) and corrected radar stand precipitation at canopy level (data from DWD (Deutscher Wetterdienst, Germany), ASTA (Administration des Services techniques de l'agriculture, Luxembourg) and KNMI (Koninklijk Nederlands Meteorologisch Instituut, Netherlands) derived after Neuper and Ehret, 2019). The interception in the canopy and litter layer is not addressed. There is no understory vegetation at both sites.

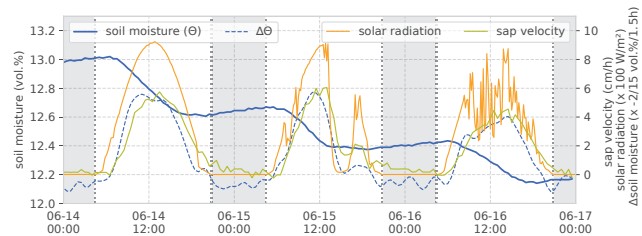

**Figure 3.** Example of observed soil moisture, sap velocity and solar radiation during three days in the vegetation period. The example is from the sand site dataset, soil moisture values are in 0.7 m depth. June, 14 is a sunny day with clear sky conditions, June 15 has clear sky intermitted by one shading spell in the afternoon and June 16 is a day with fair weather and radiation noise by scattered cumulus clouds. Shaded areas refer to astronomical night time. Notice that the change in soil moisture is inverted in the plot for easier comparison to sap velocity and radiation.

## 3 Methods

Estimating RWU from changes in soil water is not a novel idea in general (Novák, 1987; Feddes and van Dam, 2005). With precise and distributed measurements, step-like dynamics of soil moisture are observed on days with negligible vertical soil water movement (Guderle and Hildebrandt, 2015). These steps coincide – and the respective soil moisture changes highly cor-
relate – with the observed sap velocity dynamics. For illustration, we selected an exemplary three-day interval in the vegetation period. This interval contains a sunny day with clear sky conditions, a day with clear sky intermitted by one shading spell and a day with fair weather and radiation noise by scattered cumulus clouds (Fig. 3). The correlation between changes in soil moisture and sap velocity give a Spearman rank correlation ($r_s$) of 0.87. Applying the Kling-Gupta-Efficiency (KGE) which considers the contributions of mean, variance and correlation when calculating time series deviations (and is thus sensitive to
both the curve shape and its absolute values) yields a value of 0.64 (after linear scaling of the value ranges). Especially on June 15 the coherence between solar radiation, sap velocity and change in soil moisture becomes very obvious, when intermittent cloudiness lets radiation and sap velocity drop in the afternoon. During the same period the decline of soil moisture is halted, too. Furthermore, one can see that the signal of sap velocity follows the solar radiation with a slight time lag. Change in soil moisture follows the same pattern. When we can exclude percolation and pedophysical soil water redistribution as main drivers
of soil moisture change, we may attribute these observed steps in the rhizosphere soil water content to RWU. The remainder of this methods section explains the steps to estimate daily RWU and SF (as illustrated in Fig. 4 and Fig. 5) and our approach of a comparison of both fluxes.

### 3.1 RWU calculation

Based on the idea of Guderle and Hildebrandt (2015) and Blume et al. (2016), we developed an algorithm to identify the
characteristic declines and to extract daily RWU from the observed differences of soil water between two sunsets (Fig. 4 A).

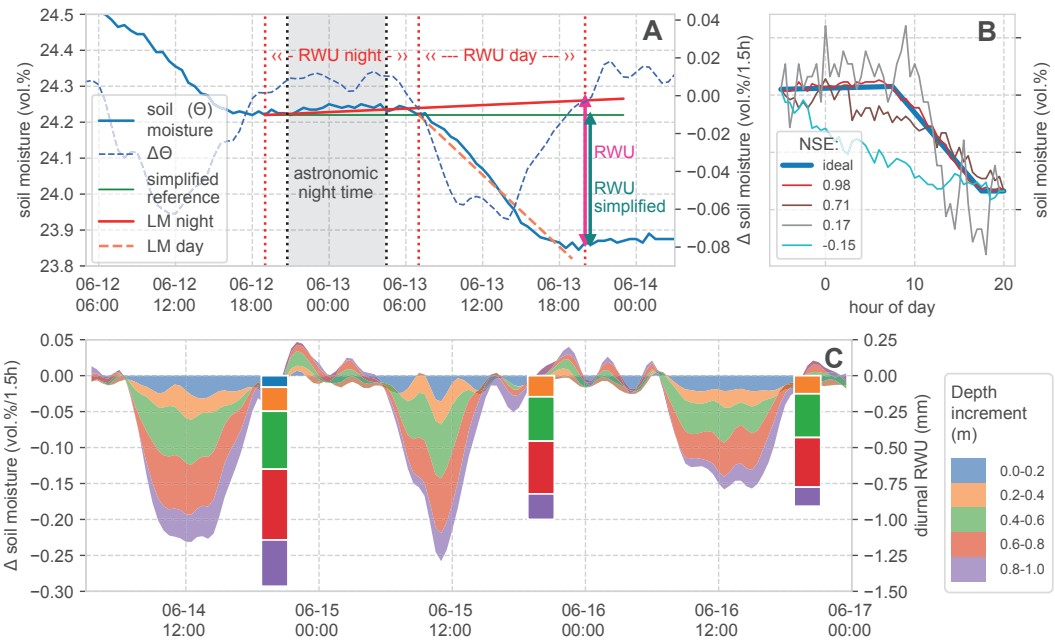

**Figure 4.** Calculation of root water uptake from soil moisture change. **A:** Time series of soil moisture and change of soil moisture during one day in one soil layer and indication of calculated RWU, showing the effect of including a linear regression model for nightly water redistribution (LM night) compared to the simplified calculation. **B:** Comparison of several exemplary (scaled) soil moisture declines to an artificial reference step ("ideal") and calculation of the Nash-Sutcliffe-Efficiency (NSE) for evaluation. **C:** Stacked change in soil moisture in the top soil layers and calculated daily root water uptake (bars) with the top measured soil layers also at the top of stack. No RWU is calculated if the time series does not meet the required basic criteria of the step shape (presented in section 3.1). This was not the case for the 0–0.2 m layer on June 15 and 16, hence the blue color at the top of the stack is missing for these days. The stacked bars form the basis for the comparison in Fig. 6.

First, we identify the inflection points of the time series (Fig. 4 A, vertical dashed red lines). These points are i) the beginning of a decline of soil moisture after sunrise and ii) the end of this decline near sunset. The astronomic reference times have been calculated with the *Astral* package (Kennedy, Simon, 2019) using the geographic positions of the sites. Our algorithm scans for the first soil moisture change of $\geq 0$ vol % h$^{-1}$ in a window starting five hours before sunset and identifies this as the beginning of the night. The next decrease below $-0.02$ vol % h$^{-1}$ is marked as beginning of diurnal RWU. The beginning of the next night is used as final evaluation reference. This approach is sensitive to noise in the data. Due to the high quality of the employed TDR sensors we could avoid strong smoothing. To make the procedure more robust, we applied a 1D Gaussian filter with one standard deviation to the resulting time series of changes in soil moisture before evaluation.

Generally, the estimate of the diurnal RWU is simply the reduction in soil moisture between two days (Fig. 4 A, green line). We extend this simplified approach, to account for hydraulic redistribution of soil water in the rhizosphere. We assume that such redistribution fluxes manifest as changes in soil moisture during the night but remain active during the day. To calculate

these changes we fit a linear regression model (LM) to the observed soil moisture time series during the night and extend it to the reference time at the end of the day (Fig. 4 A, slightly increasing red line). Now, the calculated difference in soil moisture compensates for hydraulic redistribution. In the time series in our example, soil moisture is increasing during the night. There are also cases with slightly decreasing nocturnal soil moisture. We stick to the approach correcting for hydraulic redistribution

in the following analyses and later evaluate its benefits compared to the simplified version.

Because diurnal change in soil moisture is not necessarily RWU, we assess a) the general step shape of the observed daily declines and b) the occurrence of external fluxes which could dominate soil moisture changes before estimating RWU. To this end, we calculate the slope of linear models (LM) fitted to both night- and day-time changes in soil moisture respectively (Fig. 4 A). We define the following criteria to characterise the expected step-shape:

**a)** The day-time slope of soil moisture is negative (decline in soil moisture during the day) and three times smaller than the night-time slope (general step shape of the curve).

**b)** The night-time slope of soil moisture remains at moderate levels of diffusive flux rates between $-0.01\,\text{vol}\,\%$ and $0.02\,\text{vol}\,\%$ in $12\,\text{h}$. A stronger decline in soil moisture during the night would indicate percolation or external withdrawal as a dominating process, whereas a larger increase would indicate an external input of soil water.

In the identified steps which meet the given criteria, the change in soil water content over the day is calculated at the beginning of the next night period (Fig. 4 A, magenta and green vertical arrows). Fig. 4 C gives an example of the resulting daily RWU estimate for the first metre of the soil profile, alongside the corresponding changes in soil moisture. There, one can also see that on June 15 and 16 the soil moisture dynamics in $0$–$0.2\,\text{m}$ depth did not meet the criteria for the step shape. Hence there is no RWU estimate in this layer.

## 3.2   Evaluation of the estimated RWU

In addition to the general check of the step shape of soil moisture dynamics during the calculation of RWU, we add an evaluation measure of how well the observed diurnal step agrees with a synthetic reference.

For this, we construct a synthetic, "ideal" step based on the observed soil moisture values at two successive sunsets and our criteria for the expected step shape (see Sec. 3.1). Between the observed values at sunset, we insert an increased moisture

value (by $0.01\,\text{vol}\,\%$) at $3\,\text{h}$ past astronomic sunrise and let the value at sunset be reached $3\,\text{h}$ early. The intermediate values are linearly interpolated (Fig. 4 B, blue line). This synthetic reference is compared to the observed time series by calculating the Nash-Sutcliffe-Efficiency (NSE). The NSE is a measure which is very sensitive to deviations from shape features.

Fig. 4 B contains several observed soil moisture steps and their respective NSE values. For all steps, the general criteria are met but the deviations from the idealised step can be quite substantial. This can be due to signal noise or due to other

reasons causing a reduction in soil moisture. We expect an NSE $\geq 0.5$ to be a fair reference for good agreement of the observed dynamics with mainly RWU-driven soil moisture decline.

As a qualitative evaluation, we compare the number of detected steps in each soil layer with the total number of days with $\text{SF} > 0.1\,\text{L}\,\text{d}^{-1}$.

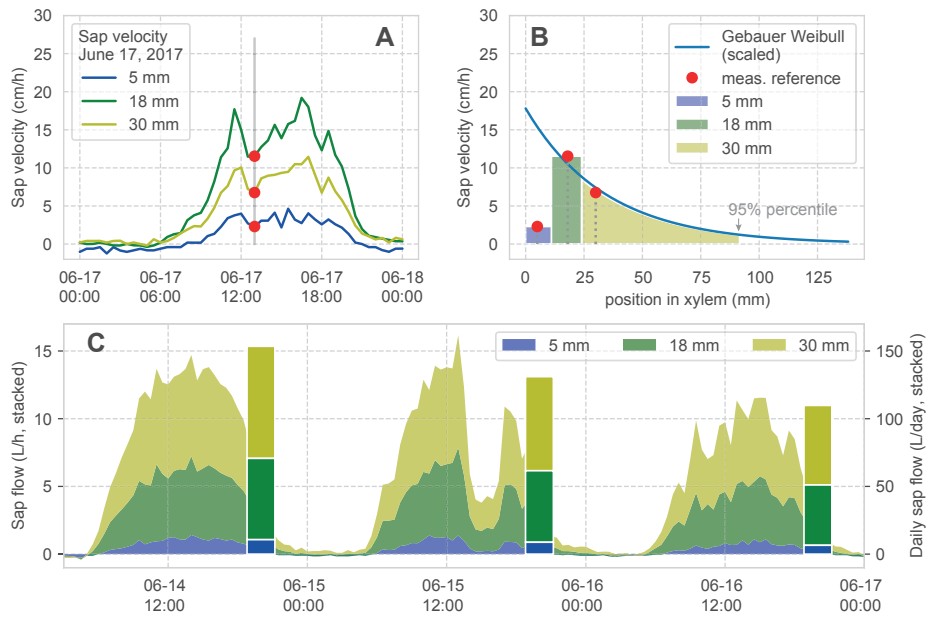

**Figure 5.** Calculation of sap flow. **A:** Measured sap velocity dynamics on June 14, 2017 for the three measurement points of one sensor, **B:** Fit of a Weibull distribution after Gebauer to the measured reference sap velocities in A. This is required to estimate the radial velocity distribution and especially the border to the inactive xylem (95% percentile of the distribution) for the calculation of sap flow. The width of the coloured bars and shaded area under the curve show the three respective increments which are used for the calculation of the sapwood area corresponding to each velocity measurement. **C:** Stacked time series of calculated sap flow and its daily aggregate (bars) for a hypothetical tree with a DBH of 64 cm for three consecutive days. The stacked bars form the basis for the comparison in Fig. 6.

### 3.3 Conversion of sap velocity to volumetric flux rates

After processing the original heat pulse measurements (Hassler et al., 2018), we obtain sap velocity observations at three positions (5 mm, 18 mm and 30 mm) within the tree xylem, measured from the cambium (Fig. 5 A). Calculating sap flow from the individual velocities requires multiplication with the corresponding sapwood area for each measurement point. Moreover, one needs to consider that i) the areas of the respective sapwood increments corresponding to each velocity measurement are dependent on the tree DBH, and that ii) the sap velocity in the xylem is unevenly distributed over the sapwood area (Gebauer et al., 2008). Ignoring this can lead to strongly erroneous estimates (Čermák et al., 2004).

We calculate the three sapwood area increments corresponding to our measurements, based on the measured DBH and the position of the sensors. Since our sensors are positioned directly in the xylem but DBH includes bark, we removed the bark thickness from our xylem area for further calculations, after Rössler (2008). The two outer sap velocity measurement points are considered representative for the radial area between 0–11 mm and 11–24 mm, respectively. These depths are the mid points between the sensor positions within the xylem measured from the cambium (Fig. 5 B). For the inner part of the active xylem radial sap velocity profiles have been shown to follow a Weibull distribution (Gebauer et al., 2008). We fit this distribution

with the parameters for beech (Gebauer et al., 2008) to the observed measurements at $18\,\text{mm}$ and $30\,\text{mm}$ for each time step, via a scaling factor (Fig. 5 B). The transition from active to inactive sapwood is determined with the 95% percentile of the Weibull distribution (Gebauer et al., 2008), which finally defines the required integral for the third sapwood area increment. The resulting time series is now reporting SF in $\text{L}\,\text{h}^{-1}$ and is aggregated to daily values. Fig. 5 C shows the stacked time series

for our example period and the daily aggregated stacked bars, which we use in the forthcoming analyses.

### 3.4    Estimation of RWU as volumetric flux

In order to rigorously compare the signals of RWU in the rhizosphere and sap velocity in the tree stem, we refer to the respective volumetric fluxes. We have already converted the observed sap velocity (given in length per time) to SF (given in volume per time). RWU (given in change in soil moisture per time) then needs to be converted into a volumetric integral as well. We

evaluate the validity of our RWU approach based on a closed diurnal water balance assuming that water storage in the tree stem has a minor effect.

    With RWU as withdrawn soil moisture in increments of $0.2\,\text{m}$ over a continuous profile, we are basically left with a guess about the lateral dimensions of the rhizosphere to derive a flux. This lateral extent can be estimated as a specific area, which is the scaling factor of a linear regression of sap flux ($\text{L}\,\text{d}^{-1}$) and RWU ($\text{mm}\,\text{d}^{-1}$) with zero intercept.

As most simple assumption, we consider the rhizosphere to be cylindrical – although it is known that the shape is highly species and site specific (Kutschera and Lichtenegger, 2002). This allows us to convert the lateral reference area into the mean rhizosphere radius as further evaluation reference for the proposed approach.

### 3.5    Comparison of RWU and SF

The quantitative comparison of derived RWU and SF is based on the calculated volumetric fluxes. As validation of our RWU

calculation and with respect to our second research question, we evaluate the correlation between RWU and SF at the two sites. For this we use the Spearman rank correlation and the Kling-Gupta-Efficency (KGE). KGE is sensitive to both the curve shape and its absolute values by considering mean, variance and correlation of two time series. In addition to evaluations of the full time series, we apply the measures in a moving window of 21 days, to account for the non-uniformity of the processes over the vegetation period.

For an analysis of the effect of soil and site characteristics on RWU and SF (third research question), we compare SF and RWU between the two sites using the same methods.

    Finally, we calculate all correlation measures also for the simplified RWU method as final check-up if including hydraulic redistribution in our method holds any merit.

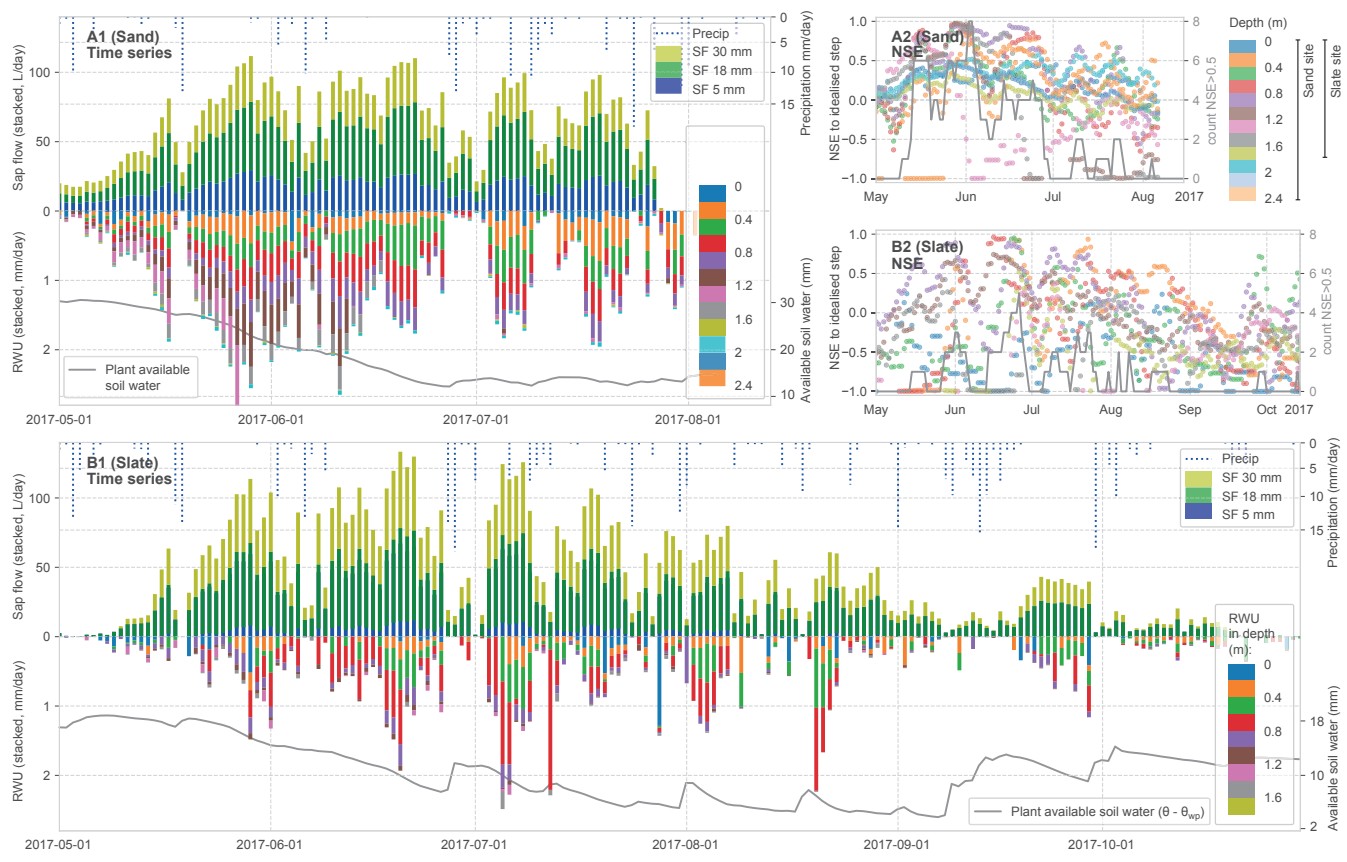

**Figure 6.** Summary of calculated time series for SF and RWU estimates as stacked daily values (see Fig 4 and 5 for methods). SF (upper half) is given as volume flux, while RWU (lower half) is given as flow of withdrawn water (without an assumption of the lateral dimension of each soil layer tapped by roots). As an indicator for soil moisture state, we report the plant available soil water in the soil column as the difference between the measured water content minus the water content at the permanent wilting point (grey line a the bottom of the panels A1 and B1). In panels A2 and B2 the evaluation of the observed diurnal soil moisture time series to the idealised step is reported for each soil layer at the two sites (rolling 7-day mean of NSE to avoid scatter) alongside the daily counts of detected steps with NSE > 0.5 across all layers. High NSE values point to high determination of the RWU estimates in the stacked bars in A1 and B1.

## 4 Results

### 4.1 RWU calculation

Building on the pre-processing leading to the stacked bars in Fig. 4 C and 5 C, Fig. 6 presents time series of daily SF and estimated RWU for both sites. The top half of each panel shows stacked daily SF and precipitation, whereas in the respective lower halves the stacked RWU estimate from the different soil layers is displayed. As an indicator for plant available soil water, we accumulated the soil moisture above the permanent wilting point over the soil profile. At the sand site, two summer thunderstorms damaged the loggers in the middle of the vegetation period, which caused an early end of the time series.

Water transport activity in the SF time series is linked to radiative forcing: during days with observed precipitation, a respective drop in SF can be seen. The general decline in tree water fluxes over the summer appears to be halted with a rain spell in mid September and higher activity in a subsequent sunny spell.

The RWU identified from change in soil water content follows the course of SF over the year, which is seen as general symmetry along the time axis in Fig. 6 (panels A1 and B1). It starts with leaf-out and increasing water fluxes through the tree until end of May. In July, both fluxes start to decrease again. In later summer with less plant available soil water, several days do not show a RWU signal although the SF signal continues at lower rates. Similarly, the evaluation of the coherence of the diurnal soil moisture steps with a synthetic step as NSE follows this seasonal pattern with decreasing compliance later in the year (Fig. 6 panels A2 and B2). A substantial proportion of the identified steps scores below the intended reference NSE value of 0.5.

Fig. 6 panels A1 and B1 suggest that the depths of RWU and the magnitude of the sourcing for each depth are not static over the vegetation period. During leaf-out both sites show RWU from deeper layers. Especially at the sand site, the sourcing from below $1\,\mathrm{m}$ depth can only be found before mid July. But also intermediate soil horizons appear to disconnect over time. It is interesting to note that the two sites differ mainly in the contributions from the shallow and deeper layers. The frequent occurrence of low NSE values of the identified step shape (Fig. 6 panels A2 and B2) suggest that the method reaches its limits not only when RWU is insignificantly small (such as in earlier spring and autumn), but also when soils are dry (most prominently between July and September). However, the count of detected steps with an NSE > 0.5 (Fig. 6 panels A2 and B2, grey lines) is not entirely explained by plant available soil water. It remains difficult to discern the interlaced effects causing the seasonal pattern within the scope of this study.

### 4.2 Comparison of RWU detection and sourcing to SF

Following our approach to evaluate the RWU detection against the occurrence of SF > $0.1\,\mathrm{L\,d^{-1}}$, Fig. 7 reports the number of days with successful RWU detection in relation to days with SF. In order to set this binary, qualitative measure into perspective, we included the total sum of detected RWU for each layer in the plot (Fig. 7, red bars).

In the most active part of the rhizosphere ($0.2$–$1\,\mathrm{m}$ at both sites), RWU was detected in about $80\%$ of the SF days at the sand site and in about $60\%$ of the SF days at the slate site. In general, a large proportion of steps could be identified with acceptable certainty (NSE > 0.5) in the most active layers. However, there remains substantial uncertainty about the step shape at both

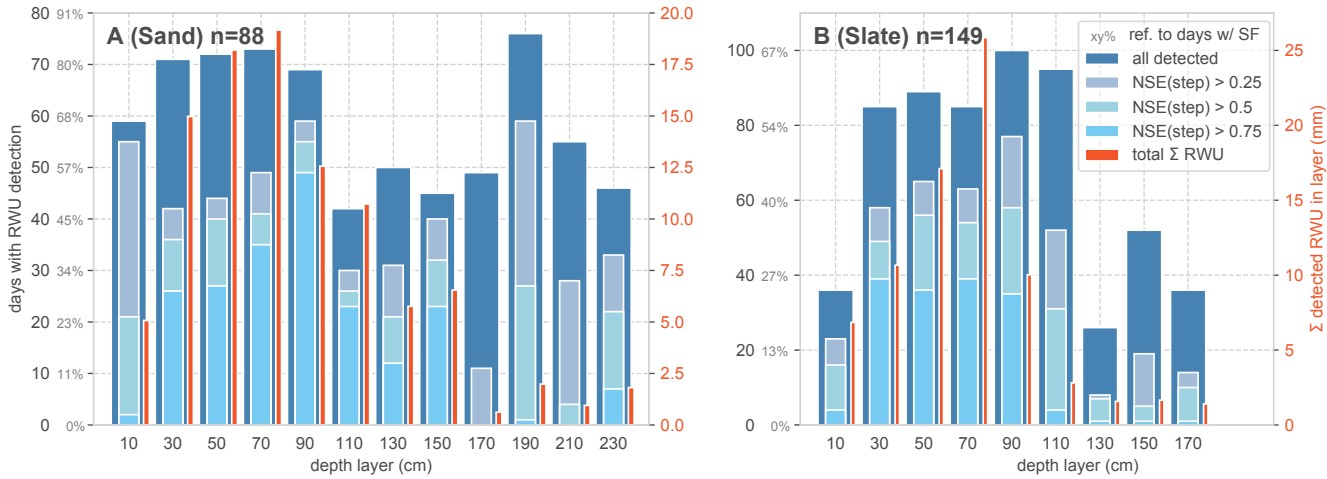

**Figure 7.** Qualitative evaluation of the RWU derivation algorithm for the sand (A) and the slate site (B). The reference n is the number of days with a SF > $0.1\,\mathrm{L\,d^{-1}}$. The dark blue bars refer to the number of days with a successful RWU detection according to the general step criteria. Their proportion of the reference SF days is given in the grey percentages on the y-axes. The lighter blue colours report the compliance of the detected RWU with the ideal step-shape assessed with the NSE, as respective sub-fraction of the total dark blue bar height. For comparison to the magnitude of the detected RWU in each layer, the total sum of RWU over the entire season is given as red bars.

sites. The overall detection rate and the step compliance are better at the sand site. A further reference about RWU magnitude and step shape NSE is given in the Appendix Fig. B1.

Comparing the depth distribution of total RWU sourcing at both sites (Fig. 7, red bars), the sand site appears to supply the water more evenly distributed over a larger range of the rhizosphere down to $1.5\,\mathrm{m}$. The slate site strongly peaks in $0.7\,\mathrm{m}$

depth and appears to deliver little water supply from below $0.9\,\mathrm{m}$. However given the limits of representative soil moisture measurements in structured soil settings, this might be an artefact of the method.

### 4.3   Comparison of seasonal RWU and SF dynamics

The sites differ strongly in the dynamic pattern of RWU sourcing (Fig. 6 and Appendix Fig. B1). In sand the tree sources water from deeper layers during spring and early summer. This deep RWU ceases over the course of the vegetation period although

overall soil moisture decreases only slightly. Such deep sources were not detected at the slate site. However, we cannot exclude that roots may source water from the weathered bedrock below the reach of our soil moisture sensors.

Looking at the correlation of the RWU estimate and SF (Fig. 8), the sand site presents constantly higher RWU/SF ratios during the onset of the growing period compared to summer. However, with an $R^2$ of 0.91 the correlation of both signals is quite high. At the slate site, the correlation is less well-determined ($R^2$ of 0.72). Despite the larger scatter, the correlation

appears to be influenced by the deviating values in the second half of the vegetation period, which are not included in the sand site data.

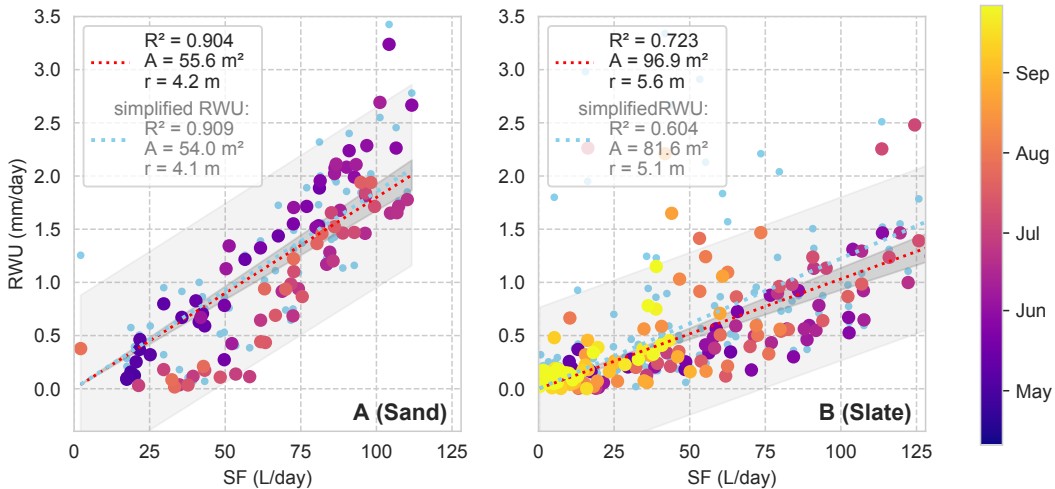

**Figure 8.** Daily RWU in relation to SF for both sites. The colour coding corresponds to the day of the year. Linear regression models are given as dashed lines. The grey shades show predicted and observed confidence intervals. The linear regression model is assumed with zero intercept resulting in a scaling factor which is reported as mean area (A) and radius (r) of a cylindrical rhizosphere in the legend (possible storage in the tree is neglected). The light blue dots and regressions refer to RWU estimates with the simplified RWU calculation approach for comparison.

Based on the assumed closed daily water balance between SF and RWU, we can calculate an estimate of the mean rhizosphere radius from the linear regression (Fig. 8). At the sand site the cylinder would have a radius of $4.2$ m. At the slate site one would estimate a radius of $5.6$ m. We use these values to calculate the fluxes for the following correlations.

The temporal dynamics of the estimate of RWU and observed SF correlate quite well with an overall Spearman rank corre-
lation coefficient of 0.89 and 0.76 for sand and slate, respectively (Fig. 9). However, the high initial correlation drops in July. At the sand site, this marks the shift to RWU ranging below SF. At the slate site, no such transition is apparent, but correlation decreases with decreasing plant available soil water. The KGE hints to slightly lower correlation of the exact dynamics and flow volumes (0.62 and 0.56 for sand and slate). Both measures corroborate the visual findings in Fig. 6 that the correlation in summer (between July and September) is less convincing. While this might be a limit of our RWU estimate, it can also point
to limitations of our working hypothesis of a closed water balance between RWU and SF.

A comparison between the two sites (Fig. 10) clearly depicts a very high correlation of SF (Spearman rho of 0.94 and KGE of 0.64) compared to weaker correlation of RWU (Spearman rho of 0.52 and KGE of 0.3). It is interesting to note that the correlation of SF remains almost constant over the whole period, while the RWU correlation is more dynamic. As we would assume a constant influence if this variability would result from an artefact of our method, the differences point towards
contrasts of the RWU process between the sites.

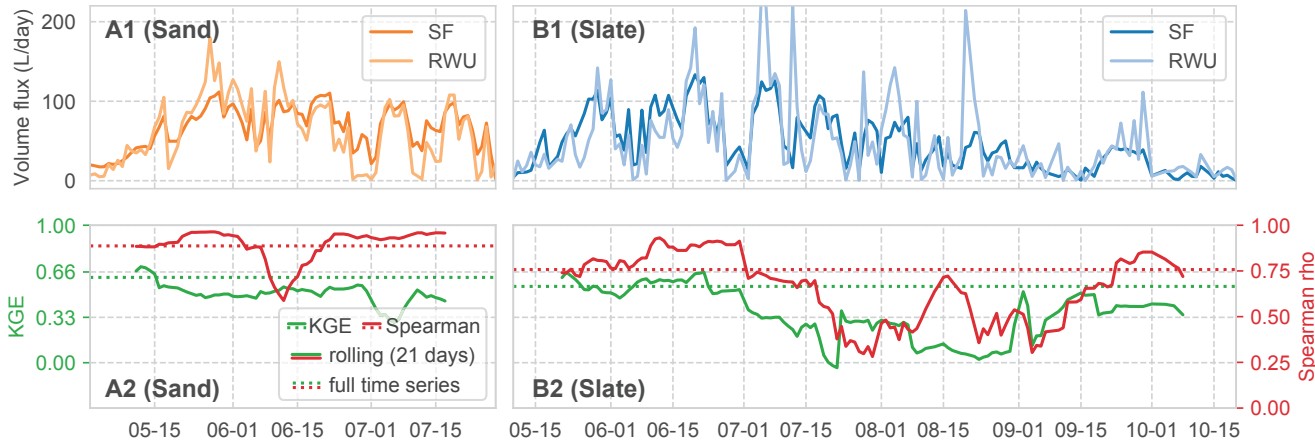

**Figure 9.** Comparison of the time series of calculated volume fluxes for RWU and SF, at both sites (panels A1 and B1 for sand and slate, respectively). Correlations between the RWU and SF time series are shown in panels A2 and B2, both as KGE and Spearman rho. The solid lines for the correlations show a 21-day rolling mean, the dashed lines are the mean correlations for the whole time series.

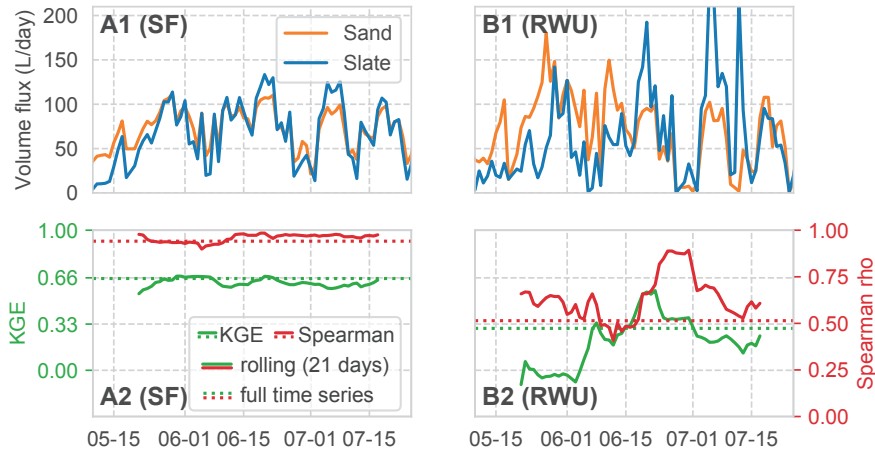

**Figure 10.** Comparison of the time series of the calculated volume fluxes for RWU and SF, between both sites (panels A1 and B1 for SF and RWU, respectively). Correlation of the fluxes between the sites, of RWU and SF between both sites, both as KGE and Spearman rho. Signatures are similar to Fig. 9.

## 4.4 Evaluating the benefit of including nocturnal water redistribution in RWU calculation

We employed the more sophisticated approach of determining RWU including potential nightly recharge via a linear regression (as described in Sec. 3.1 and shown in Fig. 4A). To evaluate whether we gained any improvement in the RWU estimates from this method compared to the simplified approach, we consider the general correlation (Fig. 8, blue signatures) and repeat the previous comparison with the simplified approach.

In Fig. 8 we see that our approach does not show any substantial effect for the sand site, compared to the simplified approach. However, at the slate site, our method appears to improve the estimates more substantially (improved $R^2$ from 0.60 to 0.72). The effect of using our approach over the simplified one on the correlations between RWU and SF was negligible for the sand site: Spearman $r_s$ improved from 0.85 to 0.89, KGE decreased from 0.66 to 0.62. At the slate site, the overall improvement of the Spearman $r_s$ is from 0.67 to 0.76 points, for KGE from 0.38 to 0.56 when applying our approach. In accordance with the observed temporal differences in the correlation (Fig. 9B2), phases of improved and decreased correlation exist using the more sophisticated approach.

Comparing RWU correlation between the two sites also shows this improvement as an increase in Spearman's rho form 0.42 to 0.52 when using our approach. However, KGE remains almost the same with 0.27 increasing to 0.30, which we attribute to the observed differences of the RWU dynamics between the sites in general.

Overall, this points to an improvement of RWU estimates when including nightly recharge especially for sites with more heterogenous soil conditions.

## 5 Discussion

Our results give a nuanced picture: Inferring RWU from changes in soil moisture within the rhizosphere is possible with our approach and provides interesting insights into the hydrological functioning of the root system for individual sites. The relatively high temporal resolution of the data, its continuous spatial distribution and its quality enables a perspective into the rhizosphere water dynamics, which is often conceptualised in models (Kuhlmann et al., 2012) but rarely measured. At the same time, our results point to considerable limitations of the approach with respect soil water state (less detectable signals during periods of low plant available soil water), physiological state of the tree (overall seasonal pattern of the signal) and soil properties (less determination in heterogeneous soil profiles).

### 5.1 Performance of the RWU derivation algorithm

The RWU derivation algorithm appears to perform very well in general (detection of 80% in sand, 60% in salte) and can be used to evaluate a broad range of diurnal changes in soil moisture (Fig. 7). RWU is prone to be underestimated on days when the detection failed due to incoherence with our criteria. This might be the case during days with active percolation (e.g. sand site end of June, Fig. 6). We find a failure of RWU detection primarily at the slate site, visible in days with similar sap velocity

(e.g. July 5 and 6 compared to July 7 and 8) lacking the RWU signal from a usually active layer (mostly $0.7\,\mathrm{m}$). Furthermore, quite a number of RWU estimates show uncertainty about the step shape (NSE < 0.5 in Fig. 6 panel A2 and B2).

In our analyses, we neglect the contribution from direct soil evaporation or understory transpiration and focus on the tree transpiration. At our sites, understory vegetation is mostly absent and we have a characteristic thick litter layer of beech leaves. We therefore regard the effect of understory transpiration as minor. However, it is noteworthy that the performance of our RWU derivation algorithm is comparably poor in the top layer, which might be exactly due to direct evaporation from the soil.

From a more technical point of view, we had the advantage of very little noise in the measured soil moisture data with precise detection of changes in the range of 0.1 permille volumetric water content. The performance of the approach is likely to decrease quickly, when the step functions are more difficult to analyse in more noisy data in different settings and with different sensors. Moreover, we had the advantage of the (vertical) coverage of the whole rhizosphere with the used tube probes. Using more common, buriable probes at specific sample locations might have more difficulties to cover the vertical distribution of RWU (Fig. 7, red bars).

The analyses of the temporal dynamics and the differences between the two sites (Fig. 9) hint at conceptual limits of our approach and experimental design: Under somewhat ideal conditions with soil moisture sensors and roots in good contact with a rather homogeneous soil matrix and sufficient soil water availability, the diurnal steps are identified and evaluated with great confidence. In the regosol with high gravel content at the slate site, the approach is challenged when roots may source water from local pools, at contact interfaces with rocks, or in the periglacial cover beds. Although the depth resolution is very insightful, the likely non-homogeneous rhizosphere will not be fully represented by a single soil moisture profile and neglecting lateral differences. Effects such as highly active fine roots at the newly growing root tips might be overlooked. Additionally, we greatly simplified the complex form and function of the tree root architecture (Pregitzer, 2008) in the assumption of a cylindrical, evenly utilised rhizosphere.

We can answer our first research question with the affirmation that our automated approach of deriving RWU from soil moisture declines generally works, but we have also outlined its limitations. We hope to have contributed an utilisable implementation of RWU detection for further applications (Jackisch, 2019), extending the works of Feddes and van Dam (2005); Guderle and Hildebrandt (2015).

## 5.2 Correlation of RWU and SF

Scaling the sap velocities to sap flow includes many assumptions and uncertainties (Wullschleger and King, 2000; Gebauer et al., 2008). Our estimates for daily SF of the two trees range around $65\,\mathrm{L\,d^{-1}}$ at the sand site (24–99 $\mathrm{L\,d^{-1}}$ as 0.1 and 0.9 percentiles) and around $50\,\mathrm{L\,d^{-1}}$ at the slate site (7–103 $\mathrm{L\,d^{-1}}$ as 0.1 and 0.9 percentiles, days with SF $\leq 0.1\,\mathrm{L\,d^{-1}}$ were omitted). These values are within the range of results from other studies on beech trees, such as the $60\,\mathrm{L\,d^{-1}}$ (3–238 $\mathrm{L\,d^{-1}}$ as 0.1 and 0.9 percentiles) reported from 39 trees in the same area in Luxembourg (Hassler et al., 2018), the 36–370 $\mathrm{L\,d^{-1}}$ reported in a study in Slovakia (Střelcová et al., 2002) and 32–54 $\mathrm{L\,d^{-1}}$ for a study in central Germany (Kocher et al., 2013). Of course these numbers vary with respect to DBH of the trees, measurement and scaling method and the monitoring time of year, but the range of SF we calculated for our trees seems plausible.

For the quantitative comparison, we greatly simplified the rooting system by using a cylindrical shape whereas a decrease in rooting density with depth might be more appropriate (Leuschner et al., 2001; Volkmann et al., 2016). This does not necessarily entail proportional RWU as trees can adapt the uptake and transport velocity, for example to use water from moist layers even when there are less roots than in drier layers (Dubbert and Werner, 2019). However, we refrain from assumptions about the detailed processes and adaptations of the root systems and use the rhizosphere scaling as an approach to roughly estimate the corresponding water flux from RWU. As the lateral dimensions of our assumed rooting zone of 4.5 m and 5.6 m seem reasonable for beech trees (Kutschera and Lichtenegger, 2002; Lang et al., 2010; Kodrík and Kodrík, 2019), we consider this a feasible approach for our purpose.

Advancing means to monitor dynamic processes in the soil-plant-atmosphere continuum is one of the overarching aims of this study. Although soil moisture-derived RWU and SF generally correlate quite well (Fig. 8), they are not interchangeable measures for estimating transpiration. The analysis of the temporal development of their correlation (Fig. 9) supports this notion. We thus argue that observing the plant system at different gauges (RWU, SF, stem storage, leaf-level transpiration) provides the chance to actually analyse the underlying processes. This might help to answer the questions: Why is there a shift of the regression between RWU and SF over time? What is the "optimisation function" of the plant's RWU sourcing (e.g. Gao et al., 2014) and SF variability (Saveyn et al., 2008)?

Moreover, not only the presented RWU derivation has uncertainty. Measuring SF is influenced by a response of the plant to the sensor installation, and by non-homogeneous xylem shapes and associated differences in water transport around the stem (e.g. Bieker and Rust, 2010). The regression analysis (Fig.8) shows the seasonal changes in the observed flux rates. In order to further study effects of seasonal storage, different sourcing, adaptation to environmental conditions and methodological concerns, the correlation between RWU inferred from soil moisture dynamics and SF appears to be an interesting means. However, our working-hypothesis of a closed water balance between RWU and SF remains subject to further research. The observed scatter and seasonal changes in Fig. 8 hint to such effects. Further studies could benefit from measuring RWU and SF complementarily in order to gain more knowledge on the various influences and temporal dynamics of this correlation.

With regard to our second research question we do not see a consistent relation between RWU and SF. This might be partly attributed to the algorithm performance, but also indicates that RWU and SF are not interchangeable but complementary measures.

### 5.3 Effects of the sites and controls for RWU

Despite good general agreement of the SF and RWU both signals show substantial differences over the season and between the sites (Fig. 6). We have shown that the two sites have quite different RWU patterns of sourcing and temporal dynamics. It is interesting to note that the main differences in RWU occur during the leaf-out phase until end of June. SF at the two sites is highly similar throughout the year. Thus, very different subsurface water states and sources result in similar fluxes in the trees.

Our study does not allow for a conclusion about adaptation and regulation of the tree water supply in the process of photosynthesis. We cannot exclude that some of the apparent differences are due to the limited capabilities of the method. Instead we intend to contribute an easily applicable method for further studies of the interplay between RWU, SF and transpiration

(Schymanski et al., 2009; Lu et al., 2020). The presented measurements may be a means to complement analyses of the links between subsurface and stand organisation (Metzger et al., 2017) and transpiration of trees (Renner et al., 2016).

With respect to the dynamic sourcing of RWU one might be tempted to relate the observed soil moisture, SF and the calculated RWU to matric potential inferred from the same data through a soil water retention function. We have done so based on measured soil characteristics and fitted van Genuchten parameters, but found physically inconclusive results (see Appendix C). The difficulties of this approach sets us back to the general concept of soil moisture, retention properties and capillary flow (Or et al., 2015; Lu, 2020). Namely, the measured soil moisture appears to underestimate the water content in the pore space near the roots. This leads to erroneous values of the matric potential. We have seen similar conceptual shortcomings in a soil water sensor comparison (Jackisch et al., 2020).

Given this finding, the conceptualisation of plant-soil-water relations as capillary concept (e.g. Janott et al., 2010) might have essential limits with respect to state observability in the rhizosphere. Regarding multiple functions and specialisations of different roots in the root system (Kerk and Sussex, 2001), the controls of RWU and resulting transpiration require more specific approaches with higher spatiotemporal resolution. This is also the case for hydraulic redistribution in the rhizosphere (Neumann and Cardon, 2012) including modifications due to root exudates (Carminati et al., 2016). At the other end of the spectrum stem flow (Liang et al., 2011) and its root-induced preferential flow extension (Johnson and Lehmann, 2016) can become essential but have been neglected in this study.

Acknowledging the limited specificity of our soil moisture-based approach, we see differences in RWU sourcing, correlation of the fluxes and their temporal dynamics, affirming the assumption of a geological and pedological influence on RWU, which we formulated in our third research question. Including contrasting site conditions in further detailed and integrated studies of ET in forests will help to untangle some of the issues of the RWU contribution.

## 5.4 Outlook

As we have shown for moderately moist conditions, an estimate of RWU from soil moisture dynamics appears reasonably robust. Applications of RWU studies based on changes in soil moisture might benefit from laterally distributed or spatially continuous monitoring. Adding this to SF measurements gauging different roots (Lott et al., 1996) and analyses of stable isotope concentration in the xylem water (Rothfuss and Javaux, 2017) could avoid overly simplistic assumptions about soil water availability and mixing. Analyses with higher temporal resolution could also elucidate further details about diurnal variations in xylem water isotopic signatures (De Deurwaerder et al., 2019). Moreover, higher spatial coverage and resolution using hydrogeophysical, quantitative measurements like time-lapse ground penetrating radar (Allroggen et al., 2017; Jackisch et al., 2017) would enable further analyses of the active rhizosphere and its geometry. Eventually, a more realistic implementation of all compartments controlling transpiration into land surface models (e.g. Kennedy et al., 2019) could support analyses of stressors and adaptability under shifting environmental conditions.

## 6 Conclusions

Inferring root water uptake (RWU) from changes in soil moisture during days without percolation is promising. We presented an automated evaluation of respective time series of soil water profile dynamics within the rhizosphere. However, the approach is not universally suitable. The more complex the pedological setting, the more uncertain the estimate becomes. High precision and low noise in soil moisture measurements are a prerequisite for the method, especially when using an automated detection of the diurnal soil moisture decline. Furthermore, monitoring the whole rhizosphere profile instead of preselected depths proved important because the sourcing of the transpiration signal changes over the year.

Our study shows that RWU and sap flow (SF) cannot be used interchangeably as estimates for transpiration. In fact they give complementary information to understand the whole process from the soil water sourcing, transport through the tree towards eventual transpiration to the atmosphere. At our sites, we observed very different patterns in RWU despite similar SF and almost identical atmospheric forcing.

Transpiration in forests is influenced by both, site conditions and plant characteristics including their site adaptations. Therefore an experimental design of field studies complementarily measuring the different aspects of transpiration is promising (e.g. RWU from different profiles within the rhizosphere, SF, stem storage and leaf-level transpiration) to gain a holistic understanding of (evapo)transpiration.

*Code and data availability.* The RWU and sap flow calculation toolbox is published as Python package on GitHub (Jackisch, 2019). The data is available via GFZ Data Services (Jackisch and Hassler, 2019).

## Appendix A: Slate site sap flow reference

At the slate site, the sap velocity measurement in the intended tree for reference failed three weeks after leaf-out (T3, DBH of $41\,\mathrm{cm}$). Hence we needed to refer to another beech tree at the site (T1, DBH of $48\,\mathrm{cm}$). The correlation of sap flow of all three monitored beech trees at the site (Figure A1) shows convincing overall signal similarity ($r_s$>0.8) but stronger deviation in absolute sap flow values (low KGE). The strongest deviation occurred in the three weeks after leaf-out. This time period also showed the strongest deviation of sap flow values between the two sites (Figure 10), due to differences in timing of leaf-out. We selected tree no. 1 (T1) to replace the intended tree no. 3 (T3) as reference based on the best correlation measures.

## Appendix B: Uncertainty of RWU calculation

We report further details about the identified RWU and the respective NSE of the step-shape (Figure B1). The almost uniform distribution of NSE values across all RWU values at the sand site indicates that there is no detection threshold for RWU. At the slate site, the distribution is skewed towards smaller RWU values. The covered range of values is the same no indication

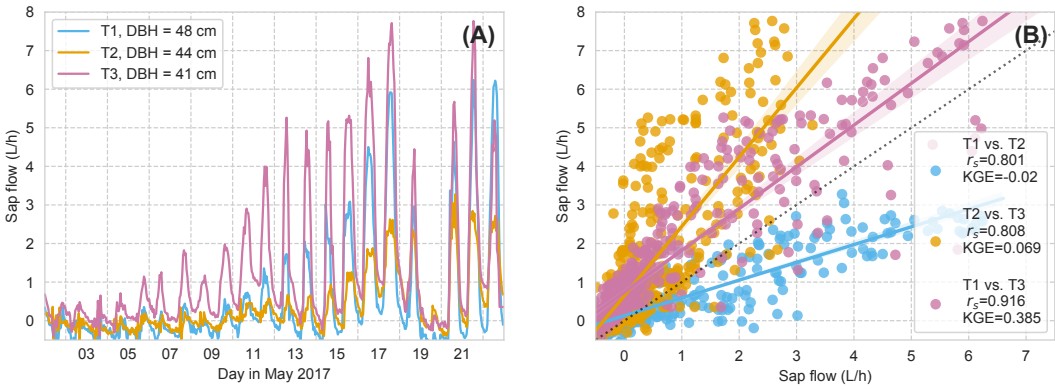

**Figure A1.** Hourly SF of all three beech trees at the slate site. (A) Time series, (B) Correlation and KGE between time series. T3 is the tree at the soil moisture profile. T1 is the tree used as reference in the study.

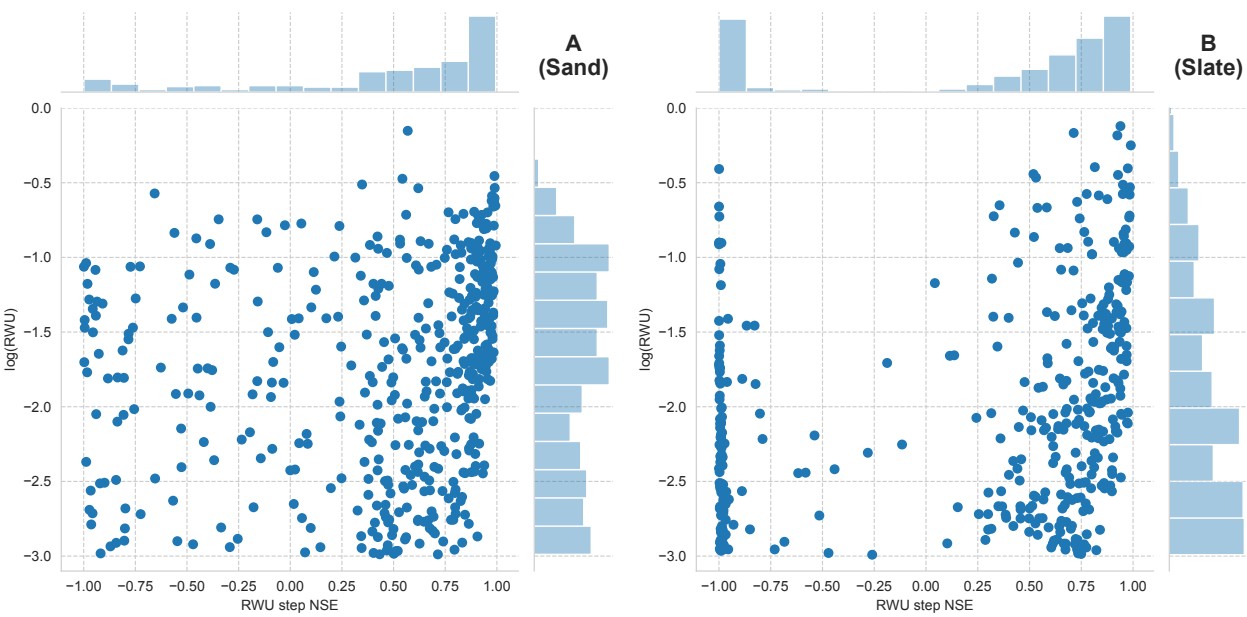

**Figure B1.** Identified RWU (log10 of RWU in mm/day on y-axes) and corresponding step coherence as NSE to synthetic step (x-axes) for the sand site (A) and slate site (B). Marginals give respective histograms.

for a detection threshold, too. At the slate site, a larger number of days have a NSE below zero, which might be false-positive results but which might also be another manifestation of the site characteristics discussed in the main part of the manuscript.

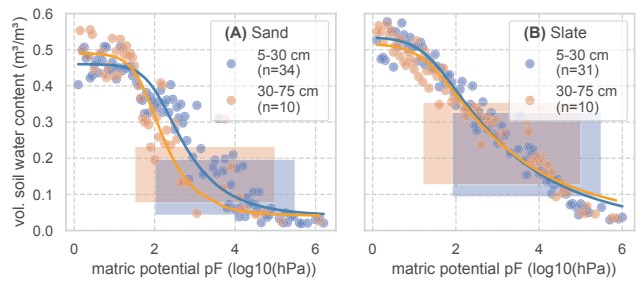

**Figure C1.** Soil water retention curves for two soil layers at both experimental sites. To derive the retention curves, the matric potential is divided into bins of 0.05 pF. Measured soil moisture values of all samples and at tensions that fall into each bin are averaged and displayed as dots. The retention curve is fitted to these points. The resulting van Genuchten parameters are given in table C1. The number of soil samples that form the basis for the retention curves is given as n. The shaded areas mark the range of soil moisture values we observed with the TDR probes in this study.

|  | Sand (5-30 cm) | Sand (30-70 cm) | Slate (5-30 cm) | Slate (30-70 cm) |
|---|---|---|---|---|
| $\theta_{sat}$ | 0.46 | 0.49 | 0.535 | 0.517 |
| $\theta_{res}$ | 0.041 | 0.041 | 0.011 | 0.028 |
| $\alpha$ | 0.84 | 1.71 | 4.13 | 4.39 |
| $n$ | 1.47 | 1.64 | 1.21 | 1.21 |
| $m$ | 0.32 | 0.39 | 0.17 | 0.17 |
| $k_{sat}$ | 7.4e-5 | 6.5e-5 | 1.92e-4 | 4.13e-4 |

**Table C1.** Table of measured soil water retention curve parameters. $\theta$ in $\mathrm{m^3\,m^{-3}}$, $\alpha$ in $\mathrm{m^{-1}}$, $k_{sat}$ in $\mathrm{m\,s^{-1}}$

## Appendix C: Soil water retention and RWU sourcing

Soil water retention properties of the soils at both sites were assessed in a previous study using the free evaporation method of the HYPROP apparatus and the chilled mirror method in the WP4C (both Meter AG) with $250\,\mathrm{ml}$ undisturbed soil samples from the sites (Jackisch, 2015). Following this method, the matric potential is divided into bins (0.05 pF). All retention data of

5  the reference soil samples is bin-wise averaged to form the basis for the fitting of a retention curve (Figure C1, parameters in table C1). We have aggregated the results of 44 and 41 soil samples in the subbasins of the sand and slate site for a more robust representation (as discussed by Loritz et al., 2017). The resulting van Genuchten parameters is given in Table C1 and Fig. C1.

When applying the identified soil water retention curve to the observed soil moisture state values, we can relate the calculated RWU to matric potential in the respective depth layer. This alternative view of the data is given in Fig. C2. Although no clear

10  correlation of RWU and matric potential can be seen, the depth-related colour coding corroborates the strong differences

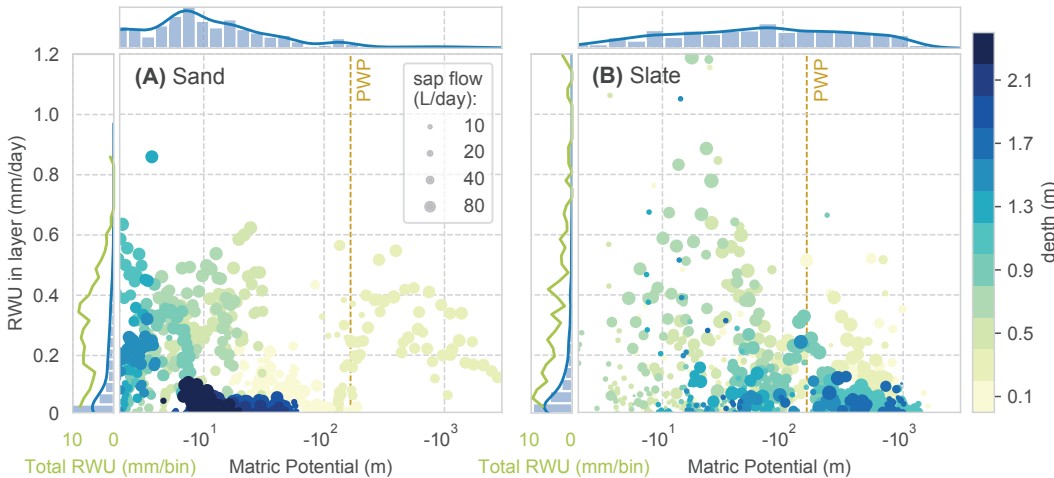

**Figure C2.** Sourcing of RWU: Daily values of matric potential in each soil layer and RWU. Colour coding with respective depth. Dot size marks the reference SF of the day. The marginal histograms and kernel density distributions on top refer to the occurrence of the respective matric potential bin in the observed period. The marginals on the left give the distribution of RWU (blue) and the total RWU of a certain bin (green). PWP marks a matric potential at the wilting point with pF 4.2.

between the sites. In general, there is more tolerance of RWU to higher matric potential at the sand site. At the slate site we recover the peak in RWU from intermediate depth (around $0.7\,\mathrm{m}$), which coincides with low matric potential.

However given high RWU rates at apparently higher tensions than the wilting point (PWP), we cannot trust this relation. Most likely this result corroborates the limits of the concept of soil moisture dynamics in structured soils. The soil water in the
5   layer is not evenly distributed and we underestimate the soil water content in the pore space which is tapped by the roots.

*Author contributions.* .

CJ and SH developed the study layout, performed the field work, prepared the data and composed the manuscript. SK did the first analyses on this data for his BSc thesis. CJ developed the detection algorithm, compiled most of the data analysis and plots with frequent discussion with SH. TB did preliminary RWU analyses based on soil moisture dynamics within the CAOS
10   research unit. The resulting discussions between TB, EZ, SH and CJ initially triggered the study. EZ and TB supportively accompanied the study and contributed during the manuscript preparation.

*Competing interests.* The authors declare no competing interests.

*Acknowledgements.* This study contributes to and greatly benefited from the "Catchments As Organized Systems" (CAOS) research unit. We sincerely thank the German Research Foundation (Deutsche Forschungsgemeinschaft, DFG) for funding (FOR 1598, ZE 533/9-1). We thank Malte Neuper for the preparation of the rainfall data and Anke Hildebrandt for inspiring discussions about this study. We sincerely thank Jesse Nippert, Leander Anderegg, Jia Hu and Chris Still for their constructive review comments which greatly improved the manuscript.

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
