# Peer review of "Estimates of tree root water uptake from soil moisture profile dynamics"

_Biogeosciences, 2019_

## Short Comment (SC1) · 14 Jan 2020

Here the authors use detailed TDR measurements at consecutive depths in the soil profile to infer changes in water uptake by several beech trees on varying substrates. The authors compile a rigorously quantitative set of complimentary data (including sap flux from target trees) to validate the fluxes derived from the soil moisture measurements. Generally, the 'root water uptake' metric derived from soil moisture was similar to sap flux estimates, particularly during periods with greater water availability and on the sandy site. The goal of this study – develop a new metric to provide detailed water-use / transpiration estimates in complex canopies – is novel, rigorous, and has tremendous potential for more detailed ecological investigations.

[Figure]

I have few major concerns or suggestion revisions for the authors to address. My most noteworthy suggestion for the authors is to simply / clarify this manuscript whenever possible. This is an extraordinarily detailed and jargon-rich manuscript that needs to be simplified. I'm a scientist that studies plant-water use (so in theory, this manuscript is directly within my field of study) and yet I found myself reading and re-reading sections to try to understand what was done, what the data means, and how the authors derived their conclusions. For many of the figures, I was never able to gain a full appreciation of what was conveyed (what exactly am I seeing in Figures 11 and 12?) and the legends were often non-descriptive (see Fig. 4 for an example). Thus, I encourage the authors to reduce the jargon, better explain the development of the RWU calculation, and simplify the figures whenever possible. The novelty and creativity of this manuscript is in the top 95%, but the clarity and delivery of the information is in the 50-60th percentile. I'm fully confident the authors can address this issue.

Other comments: The Introduction is generally sound, but there are a few items of concern. On page 1 line 17, the authors need to remove 'even over grasslands'. Grasslands are not definitely simpler ecosystems that forests, so please do not refer to them as such. On line 18, please address the 'optimise their water transport to respiration'. The optimality theory has been challenged many times since 2009, and in fact doesn't appear to be valid. In addition, why would water transport be optimized for respiration? I believe you meant assimilation. Regardless, please update.

Examples of RWU assessments in the literature are provided on page 2. While I have never read an approach as detailed as the one presented here, transpiration dynamics have been assessed by comparing sap flux with changes in soil moisture dynamics previously. Please check out Holdo and Nippert 2015 Ecology (https://doi.org/10.1890/14-1986.1) for a study comparing transpiration dynamics using sap flux, soil moisture, isotopes, and changes in canopy temperature among coexisting trees and grasses.

On page 5/line 7, the authors note that sap flux was monitored in 4 trees near the TDR probes at each site. Do subsequent sap flux data (Fig. 3, 5, 6, etc) represent a single

tree, or an interpolation across all 4 trees per site?

Page 5, line 22. 'Efficiency' is misspelled.

Section 3 describes Fig. 3, and the derivation of RWU based on soil moisture step change. How was this 3 day period selected? Can I assume it was the best 3-day period where a step change was observed during this summer? Does the lower correlation in RWU time series (and during drier times) reflect derivation of the metric during an ideal period, which then loses predictive power during mean summer soil moisture periods?

When linking RWU from TDR data to transpiration dynamics from an individual tree, how did you account for water use by the understory vegetative community? Unless there were no grasses, forbs, saplings, etc. – wouldn't these species be using water from the same soil depths which would then complicate predictions of individual tree water-use using changes in soil moisture? The methods section does not describe the understory.

On the sandy site, how did you account for capillary action within the soil profile and subsequent evaporation? Or do you have some information (which I may have missed) on evaporation rates, and the depths of the evaporative fronts from these soils?

Fig. 6 is an impressive behemoth. One item of interest to me is that the rainfall history does not appear to have any temporal synchrony with RWU or sap flux. Is this true? And if so, why not? In Fig. 7, how does these predictions of water use in mm/day compare to estimates of Beech from similar locations within the literature. It would be nice to know if these predictions fall in the range reported elsewhere.

Page 14, line 9 – I don't think 'ambivalent' is the correct term. An ambivalent picture suggests this was inconclusive research. You concluded many things, and illustrate a path forward for using soil moisture to infer plant water use.

Page 17, line 5. How do you know it's a minor effect? HR can be quite substantial in

many ecosystems.

Page 17, line 14 – How would cosmic ray measurements be appropriate here? I was under the impression that cosmic ray data pertains to the top 10 cm of the soil only.

In the Discussion section, the authors note that RWU and sap flux were not interchangeable, but were complementary. The language infers that RWU may be more desirable than sap flux for estimating transpiration under certain conditions. This seems a bit misleading to me. Sap flux directly measures a physiological process that relates to canopy transpiration, while RWU is an inferred metric. Under what conditions would the data from RWU be preferred over sap flux? Under what conditions does RWU outperform sap flux for predictions of transpiration?

---

## Referee Comment (RC2) · Jia Hu (Referee) · 31 Mar 2020

See attached pdf document as well.

In this study, the authors examined root water uptake in beech trees, along a soil moisture profile in two sites with different moisture conditions. There were some interesting findings. First, they found differences in depth of water taken up by the roots between the two sites (sandy soils use deeper water). This was particularly evident during the period of maximum transpiration rates. They also found that while RWU could be estimated from changes in soil moisture, there were also instances where transpiration was occurring, but RWU was not measured. In a comparison between the sand versus slate sites, although SF was similar throughout the year, RWU was quite different.

One of the main ways in which this manuscript can improve is to clearly discuss the reasons for comparing RWU and SF. The authors state that the aim of the study "is to evaluate the potential and limitations of the diurnal decrease of rhizosphere soil moisture measurements as an estimate for RWU in ecohydrological field studies." They make the point that RWU is an under measured observation, and that other proxies, such as changes in soil water content, are used to infer RWU. Meanwhile, sap flow sensors measure transpiration rates in trees, but because of stored water within trees, sap flow does not measure RWU uptake either. So linking RWU and SF (as mentioned in hypothesis 2) seems to be an important link. However, what wasn't clear for me is that if ET is an important metric to quantify for ecohydrological studies, what does RWU measurements provide that SF measurements don't? In other words, what additional processes related to ET do RWU measurements elucidate? I think this discussion could be enhanced more in the introduction. For example, in lines 28, the authors state, "Furthermore, spatially distributed monitoring of both RWU and soil moisture and SF could help to elucidate differences between the influence of the geological and pedological settings on water supply to transpiration and the influence of the plants themselves and their adaptations in root systems, dynamic sourcing of water and transpiration efficiency." Does this suggest that RWU influences the "geological and pedological settings on water supply to transpiration" while SF measurements assess the "influence of the plants themselves and their adaptation in roots systems...?" But SF also influences RWU, so shouldn't SF and RWU be considered in a framework that acknowledges that they influence each other?

Figure 3. Why are estimates of change in soil moisture positive if the soil moisture decrease throughout the three day period? In page 6, line 22, does "change in soil moisture" refer to "big delta theta?" If so, again, why is change in soil moisture positive? The positive values of change in soil moisture during the daytime is confusing because in Figure 4B, change in soil moisture during the daytime hours is shown as negative. In Equation 1, the authors also state that a check to evaluate the data is that " day slope of soil moisture is negative (decline in soil moisture during the day)..."
[Figure]

Page 7, line 1. Do the bolded a) and b) here refer to a subset of "soil moisture (b)" from page 6, line 15? If so, I would change "a) and b)" to "i) and ii)" as to not confuse the reader.

Page 7, line 4 and 5. No need to say "no STRONG decline in soil moisture" or "no TOO STRONG increase in soil moisture" since STRONG or TOO STRONG are quite subjective. I think that saying "no decline in soil moisture" or "no increase in soil moisture" followed by the rates of increase or decrease is sufficient.

Page 7, line 30. Why was the assumption made that measured sap flow originates in the soil moisture decrease? Could there be any storage of water in the trunks (i.e. might lags between RWU and SF exist)?

Page 7, line 31. "This is done by linear regression of daily sap flow to the sum of RWU over the soil profile with assumed zero intercept." Is the assumption again here that water from the different soil layers instantaneously feeds into the transpirational stream – in other words, there is no lag in when water is taken up by the roots and then transported to the trunk of the tree?

Page 8, Line 1. "The resulting factor is the mean reference area required to supply to observed sap flow." Is the 'factor' mentioned here the area or the volume? If RWU is summed across the different soil depths in which soil moisture is measured, how is the resulting factor estimated as area and not volume?

Page 10, Line 2. "In later summer, the RWU signal ceases although the sap flow signal continues at lower rates." In Figure 6, I don't see when this occurs across the entire instrument period.

Page 10, Line 32. "With a working-hypothesis of a closed water balance...the linear regression also results ....at the sandy site the cylinder would have a radius of 4.2m...slate site one would estimate a radius of 5.5m." I may have missed this, but how did you reach these readius values? Where is the linear regression model reported? I see that there are radius values reported in Figure 7, but how were these calculated?

Page 12, Line 2. "However, the high initial correlation drops in July. At the sand site, this marks the shift to RWU ranging below SF. At the slate site, no such transition is apparent." In Figure 8, when the spearman correlation drops, the precedes when RWU drops below sap flow. There are also instances later in July when RWU is consistently below SF but the spearman correlation ratio does not change. What does this mean?

Page 14. Line 9. I would recommend changing the work "ambivalent" to "mixed."

Page 16, line 9. "What is the optimization function of the plant's RWU sourcing and SF variability?" What do the authors mean by this? Please explain.

Page 16, line 12. Yes, wounding from sap flow sensors can indeed underestimate sap flux velocity, and non-homogenous xylem depths can influence estimates of total transpiration rates, but it seems unlikely that these effects would be most noticeable during periods when both sap flux and RWU begin to decline. The authors allude to other factors in the previous paragraph (e.g. stem storage, leaf level transpiration) that offer more likely explanations for why correlations between RWU and sap flux correlations decrease as the soils dry out.

Please also note the supplement to this comment:
https://www.biogeosciences-discuss.net/bg-2019-466/bg-2019-466-RC2-supplement.pdf

---

## Referee Comment (RC3) · Leander Anderegg (Referee) · 14 Apr 2020

In this manuscript, the authors pair soil moisture measurements that have high spatial and temporal resolution with tree sapflow measurements at two different sites to test whether such soil moisture measurements can be used to estimate daily transpiration and identify depths of root water uptake (RWU). They find promising similarities between sap flow and estimated RWU during a fairly wet period at their site with sandy soil, but worse correlations at a site with more heterogeneous soil characteristics and a time series that extended into a drier period. They also found interesting evidence for differences in the depth of RWU at the two study sites, though this is somewhat deemphasized in the text. While the estimated daily RWU uptake appears promising in some regards, they also found a confusing lack of relationship between RWU and soil

matric potential calculated from soil moisture release curves and soil water content. All told, these results suggest that the method is promising but still has some kinks to be worked out, some of the largest probably relate to spatial heterogeneity at large scales (lateral variation over meters) and fine scales (inability to infer matric potential from soil moisture release curves on nearby soil samples.

This is an interesting manuscript that presents a promising approach to estimating transpiration and RWU at high temporal scale. However, my three main concerns are:

1) The writing and figures are extremely dense and sometimes confusing/contradictory. I had to read the Results at least two times, and often had to parse out individual sentences multiple times before I could begin to follow their meaning. Some of this could be due to a difference in fields (hydrology vs the plant ecophysiology terminology that I am more familiar with), but I would recommend a considerable expansion of the Results to explain the more complicated an nuanced findings and make this interpretable by a broad audience. I have given multiple suggestions below in the 'Specific Comments' section, but would general recommend a careful edit and clarification of the most complicated sentences in the Results. I also would recommend simplifying some of the figures by breaking out aspects into multiple panels rather than layering on 4-5 different sources of information that I found almost impossible to interpret simultaneously. In particular, Fig 6 and 12 are nearly impenetrable (and Fig 11 is also quite dense).

2) The introduction oversells the novelty of monitoring soil moisture to estimate RWU dynamics and transpiration. True, the ability to monitor soil moisture with high enough precision to assess daily RWU is fairly novel and new, but people have been measuring soil moisture to estimate depths of RWU and understand transpiration budgets for decades! In fact, I would argue that gravimetric or volumentric soil moisture measurements are the original method for estimating transpiration (e.g. just to name a couple that come up with a quick google search: Denmead & Shaw (1962) "Availability of soil water to plants as affected by soil moisture content and meteorological conditions" Agronomy Journal; Novak (1987) "Estimation of soil-water extraction patterns by

roots" Agricultural Water Management). Thus, I think it is important in the Introduction to stress that it is the precision of these measurements (allowing high temporal and spatial estimation of RWU) that is interesting, not the method and theory itself.

3) I think the authors do a good job honestly discussing where their approach did not perform well, but I would both urge them to focus and structure the discussion around a coherent argument for what the key processes and attributes are that screw up these measurements (e.g. what are the 4 biggest problems, list them out, and show us how you concluded that these are what is causing the method to fail at the Slate site and in dry soils). I would also urge the authors to reconsider the framing and discussion around their 'Hypothesis 3'. It is currently framed as an open question whether tension gradients drive variation in root water uptake. And then Figure 11 is presented as evidence that this may not be the case. I think this is a misrepresentation of both where the field is at and what the confusing findings of Figure 11 represent. Plants can alter RWU via changes to root properties (changing aquaporin expression to alter root permeability) and root distribuitons, but they cannot physically fight potential gradients as the authors seem to suggest with Fig 11 and in the Discussion. Plants can ONLY extract and move water by moving it down a potential gradient, and there is no physical way the plant can be extracting and transpiring water from soils with a matric potential 10s-100s of MPa below 'permanent wilting point' ($\sim$1.5 MPa, or 4.2 (log10(hPa)). The general dogma (assuming +/- equivalent root resistances throughout the soil profile) that water uptake by roots should be proportional to the pressure difference and the root surface area/biomass should be used as a final test for the reliability of this method to estimate RWU, rather than using the data to test the dogma. In this case, I think it is painfully obvious that we have essentially no reliable way to convert water content to matric potential at the spatial and temporal scales that are relevant to these transpiration estimates. In fact, we're SO BAD at it, that it would appear that the Slate trees are extracting water from soil with a matric potential of « -10 MPa (when leaf water potential, the ultimate pressure differential driving water movement, is almost certinaly > -2 MPa). That tells me that there's a problem with the method, not the theory. However, recognizing this allows you to say something interesting about why we can't back calculate matric potential from these measurements (spatial heterogeneity in soil properties? Problems with rock fractions? Rock fractures that don't behave like soil samples used for dehydration curves?).

Specific comments: Pg 6 L11-14: Please explain a little more what you mean by 'NSE is a measure which is very sensitive to deviations from shape features" (perhaps you could add a day that does not pass this cuttof to Fig 4 to illustrate?), what cutoff of NSE you used, and how you arrived at that cutoff.

Pg 7 L8-12 and pg10 L30 and Fig 7: I am very confused about what 'corrected' means. In the Methods, I interpreted 'Corrected' to mean RWU extrapolated from the linear regression through the nightly data (magenta line in Fig 4a). But in Fig 7 the 'not corrected' values (blue points) are higher than the 'corrected' values (colored points), which tells me I'm getting confused somewhere. Please clarify this in Fig 4, and Fig 7 and the associated Results text (pg 10 L30).

Pg 7 L20-29: This paragraph about turning sap velocity into sap flux is very confusing. I did not understand it until I scrutinized Fig 5. Please rewrite/clarify. Also, in the Fig 5 legend/caption it is worth noting that the "5mm, 18mm, and 30mm" are depths from the outside of the tree (or inside of inner bark? Not sure which).

Figure 6: I had a very difficult time extracting the desired inferences from this figure. The shading (which varies per site, over time, and in different soil layers) is almost impossible to see and interpret (not to mention some of the colors become colors used for other soil depths when shaded) yet are referenced multiple times in the text. Also, the stacked bar plots make it almost impossible for me to interpret which depths are providing RWU, mostly I just take away total bar height. I would recommend 1) breaking out the information about how well the RWU estimation likely worked into another panel or method other than shading (filled versus unfilled bars/symbols, perhaps?). and 2) either finding a more holistic way of showing depth information (e.g. coloring whole bars

by the weighted average depth of RWU) or just making a different panel that showed line graphs of uptake by depth through time. In fact, I would potentially advocate for breaking out the depth of extraction information into a new figure altogether.

Pg 12 L14-15: I don't quite understand what data are being compared in this sentence "Comparing RWU correlation between the two sites, applying the nocturnal correction improves Spearman rho form 0.42 to 0.52. KGE remains almost the same with 0.27 increasing to 0.3." All data from both sites (if so, why is this a useful comparison)? Or somehow site-level averages?

Section 4.3 – I think this section is very cool, but I understood very little of the text. What does "a diffuse redistribution into the surrounding soil aggregates" mean and why can it be "seen as parallel declines...in the different depth layers"? Please explain more what "flashy transport through the macroporous soils and fill-and-spill mechanisms of subsurface pools" means, and much more importantly how this analysis influences our interpretation of the method for assessing RWU in this site. Clearly you learned something interesting and highly relevant (possibly that helps us interpret Fig 11?), but I do not understand what it is based on the current text. For instance, I have no idea what these sentences mean and how they relate to Fig 10b "Here, roots are likely to grow along joints and fractures, where event-water can be stored with little effect on the bulk soil moisture. As such, the measurements might miss parts of the active rhizosphere."

Section 4.4 – See above comments about interpretation and framing of these results. Also, the current Figure 11 is nearly impossible for me to interpret. I would recommend displaying SOME aspect of this information in multiple panels. (e.g. maybe splitting the soil columns up into three depths and displaying them as separate panels so you can color by SF). Also, the units/label on the x axis of this figure is confusing to me. And honestly, after reading the text of this section 4 times, I still don't have any idea what it means. I can't even decipher it enough to make suggestions on how to clarify it. I don't know what the referenced 'reactions' are and how I'm supposed to assess them in the

figure. Moreover, I do not at all see the 'correlation of matric potential and depth' that supposedly exists in slate site.

Pg 14 L14: This sentence "At the same time, we pointed out considerable limitations to the approach with respect to soil water state (no detectable signal during low moisture periods) and soil properties (high variability in heterogeneous soil profiles)" is the most interesting sentence of the discussion to me, but comes out of no where and needs much more explanation. In order for me to follow your train of thought, I require much more explanation. . .

Figure 12 and associated text of Section 5.1: I had an extremely hard time interpreting this figure. Please 1) remove the red bars for total extraction to new panels (two axes y-axes with different interpretations is much more than my brain can handle). 2) Explain what the NSC cutoffs indicate, and what the larger blue bar for 'all detected' is and why the inset bars for different detection thresholds do not sum to it 3) Put panel A and B on the same axis (e.g. 0%-90%) and switch the big numbers to be % of days and little numbers to be # of days. Also, how does Fig 12 show "The RWU derivation function appears to perform very well in general and can be used to evaluate a broad range of diurnal changes in soil moisture (Fig. 12)." (L1-2). Moreover, this sentence doesn't really make sense to me "Unlike the first impression in Fig. 6, the proportion of steps with higher uncertainty about the actual fit of the shape with the assumptions is higher in the slate site data, which is in line with the lower overall RWU detection there." Could you explain what you mean by "higher uncertainty about the actual fit"? Also, how "uncertainty" and rate of "overall detection" differ? Throughout this section, please be much more explicit about the site, times, and layers you are referring to when, for example, you write "Under somewhat ideal conditions with soil moisture sensors and roots in good contact with a rather homogeneous soil matrix and sufficient soil water availability, the diurnal steps are identified and evaluated with great confidence." Finally, this feels like it should be in the Results, perhaps even near Figure 1, rather than in the Discussion.

Pg 15- L5: I think it's worth explicityly mentioning the take-away from Figure C1: that flux amount is unrelated to how well the step function fits the daily soil moisture pattern.

Pg 16-L25-35: See my comments about Hypothesis 3 and Figure 11. Also, the sentences at L28 ("At the sandy site...") seem confusing and almost self contradictory to me.

――――――――――――――――――

---

## Author Comment (AC1) · 14 May 2020

**1   General reply to all referees**

We sincerely thank Jesse Nippert, Jia Hu and Leander Anderegg for their intense study of our manuscript and their constructive feedback. We clearly understand that we have to simplify some of the dense writing and figures to convey our findings more clearly. The referees made several detailed suggestions for this, along which we will organise the revisions. We will self-critically check for simplifications of jargon and clarity in our arguments.

With respect to the observed process dynamics of measured and inferred variables,

we will carefully revise the manuscript towards i) a more detailed description of the observed results, ii) more coherent argumentation lines, and iii) the limits of the presented approach. We will put specific attention to a) the conversion of sap flow velocity and rhizosphere water withdrawal to flux rates, b) the assessment of the coherence of the diurnal signal with the assumed step shape, and c) the reference to inferred matric potential. At a meta-level, we have to make sure not to overstretch the data set at hand which is basically a first reference. We hope that many more researchers will employ, test and evaluate the proposed approach to estimate RWU, which together will form a more comprehensive picture of the complementary information in RWU, SF and ET.

**2  Specific replies to the comment by Jesse Nippert**

The referees' comments are given in *italics* with our answers in regular font style.

*Here the authors use detailed TDR measurements at consecutive depths in the soil profile to infer changes in water uptake by several beech trees on varying substrates. The authors compile a rigorously quantitative set of complimentary data (including sap flux from target trees) to validate the fluxes derived from the soil moisture measurements. Generally, the 'root water uptake' metric derived from soil moisture was similar to sap flux estimates, particularly during periods with greater water availability and on the sandy site. The goal of this study - develop a new metric to provide detailed water-use / transpiration estimates in complex canopies - is novel, rigorous, and has tremendous potential for more detailed ecological investigations.*

Thank you very much for this summary in which we see our study well-understood.

*I have few major concerns or suggestion revisions for the authors to address. My most noteworthy suggestion for the authors is to simply / clarify this manuscript whenever possible. This is an extraordinarily detailed and jargon-rich manuscript that needs to be simplified. I'm a scientist that studies plant-water use (so in theory, this manuscript is directly within my field of study) and yet I found myself reading and re-reading sections to try to understand what was done, what the data means, and how the authors derived their conclusions. For many of the figures, I was never able to gain a full appreciation of what was conveyed (what exactly am I seeing in Figures 11 and 12?) and the legends were often non-descriptive (see Fig. 4 for an example). Thus, I encourage the authors to reduce the jargon, better explain the development of the RWU calculation, and simplify the figures whenever possible. The novelty and creativity of this manuscript is in the top 95%, but the clarity and delivery of the information is in the 50-60th percentile. I'm fully confident the authors can address this issue.*

As stated, we fully take this point. Our revisions will take special care to strongly simplify/clarify the language and figures.

*Other comments: The Introduction is generally sound, but there are a few items of concern. On page 1 line 17, the authors need to remove 'even over grasslands'. Grasslands are not definitely simpler ecosystems that forests, so please do not refer to them as such.*

We agree to the point that grasslands are not simpler ecosystems than forests and that this wording was not well chosen. We will remove it.

*On line 18, please address the 'optimise their water transport to respiration'. The optimality theory has been challenged many times since 2009, and in fact doesn't appear to be valid. In addition, why would water transport be optimized for respiration? I believe you meant assimilation. Regardless, please update.*
You rightly assume that we refer to assimilation. This will be updated. We also take your point that the optimality theory is under debate and that our study does not have the means to become a take on this level. We will reconsider how to make our point that a better knowledge about RWU can support the discussion.

*Examples of RWU assessments in the literature are provided on page 2. While I have never read an approach as detailed as the one presented here, transpiration dynamics have been assessed by comparing sap flux with changes in soil moisture dynamics previously. Please check out Holdo and Nippert 2015 Ecology (https://doi.org/10.1890/14-1986.1) for a study comparing transpiration dynamics using sap flux, soil moisture, isotopes, and changes in canopy temperature among coexisting trees and grasses.*

Thank you for pointing to this study.

*On page 5/line 7, the authors note that sap flux was monitored in 4 trees near the TDR probes at each site. Do subsequent sap flux data (Fig. 3, 5, 6, etc) represent a single tree, or an interpolation across all 4 trees per site?*

Yes, the respective data represents a single tree. In the paragraph P5L8ff. we seek to clarify this. We will add it more clearly that we are only using one tree per site in this paragraph.

*Page 5, line 22. 'Efficiency' is misspelled.*

Thank you. We will correct this.

*Section 3 describes Fig. 3, and the derivation of RWU based on soil moisture step change. How was this 3 day period selected? Can I assume it was the best 3-day period where a step change was observed during this summer? Does the lower corre-*

*lation in RWU time series (and during drier times) reflect derivation of the metric during an ideal period, which then loses predictive power during mean summer soil moisture periods?*

This exemplary period in Fig. 3 has been chosen arbitrarily as it combines clear sky conditions (day 1), clear sky with an intermediate shading (day 2) and a fair weather with radiation noise by smaller cumulus clouds (day 3). The step changes have been observed at many more days which we account for by calculating the NSE for each day. This is evaluated in Fig. 12. We do not compare other periods with any ideal one but we calculate i) a NSE between the data and an idealised step shape of one day and ii) a KGE between estimates of RWU and SF (Fig. 8). We will seek to clarify the respective references in the revisions.

With respect to our proposed approach, yes, it loses predictive power when the changes in soil moisture become relatively small and hence our assumption of the diurnal steps is no longer met. This changes with time and depth layer. Roughly summarised: We find steps in the data with a NSE persistently >0 between May and August and values near 1 between mid May and early July at both sites within the responsive layers. This temporal dynamics was intended to be included in Fig. 6 as shading but for sake of simplification we will remove it from this figure and explain it in greater detail elsewhere.

*When linking RWU from TDR data to transpiration dynamics from an individual tree, how did you account for water use by the understory vegetative community? Unless there were no grasses, forbs, saplings, etc. - wouldn't these species be using water from the same soil depths which would then complicate predictions of individual tree water-use using changes in soil moisture? The methods section does not describe the understory.*

Both sites did not have any understory vegetation. We will add a sentence on this. With

respect to evaporation we expect an effect on the top 20 cm. However, since the signal from this layer is only rarely evaluated as RWU we are quite confident to be correct here. From a discussion point for the proposed approach, we agree that in many applications with understory vegetation the soil moisture dynamics cannot differentiate between the different plants.

*On the sandy site, how did you account for capillary action within the soil profile and subsequent evaporation? Or do you have some information (which I may have missed) on evaporation rates, and the depths of the evaporative fronts from these soils?*

We do not have any reliable evaporation reference. As stated, we have to expect some evaporation from the top layer. One should however note that we have a litter layer of about 5-8 cm at the sites (we will add this to the site description). The surface is shaded once the leaves are out.

*Fig. 6 is an impressive behemoth. One item of interest to me is that the rainfall history does not appear to have any temporal synchrony with RWU or sap flux. Is this true? And if so, why not?*

The strongest correlation between rainfall and SF/RWU comes from the reduced irradiation during precipitation periods. At the slate site, we have a late activity end of September after a rainy period. Similarly, we can discern periods of stronger SF/RWU after rain spells over the summer. If this is a generally observable relationship remains unclear within the scope of this study. However, it would be rather plausible that the tree can source water when it is more easily available and the radiative forcing drives the "photosynthesis engine".

*In Fig. 7, how does these predictions of water use in mm/day compare to estimates of Beech from similar locations within the literature. It would be nice to know if these*

*predictions fall in the range reported elsewhere.*

We will try to give some literature values for comparison.

*Page 14, line 9 - I don't think 'ambivalent' is the correct term. An ambivalent picture suggests this was inconclusive research. You concluded many things, and illustrate a path forward for using soil moisture to infer plant water use.*

Thank you for pointing to this improper wording. We will revise it.

*Page 17, line 5. How do you know it's a minor effect? HR can be quite substantial in many ecosystems.*

We agree that hydraulic redistribution in the rhizosphere can be substantial and that especially at the sandy site we might miss important factors by neglecting it. We will rephrase this.

*Page 17, line 14 - How would cosmic ray measurements be appropriate here? I was under the impression that cosmic ray data pertains to the top 10 cm of the soil only.*

We refer to attempts of combining cosmic ray measurements with in-situ soil moisture measurements to overcome the point information towards a better spatial representation like the authors in Nguyen et al. (2019) propose. We will rephrase the sentence and stress the combination of methods to clarify.

*In the Discussion section, the authors note that RWU and sap flux were not interchangeable, but were complementary. The language infers that RWU may be more desirable than sap flux for estimating transpiration under certain conditions. This seems a bit misleading to me. Sap flux directly measures a physiological process that relates*

*to canopy transpiration, while RWU is an inferred metric. Under what conditions would the data from RWU be preferred over sap flux? Under what conditions does RWU outperform sap flux for predictions of transpiration?*

We do not think that RWU is more informative than sap flow. Certainly both have their merits and drawbacks. We aim to suggest to refer to both means as complementary gauges of a highly interlinked process. Unfortunately, SF does not directly measure the physiological process but a flow velocity over a more or less difficult to guess cross-section. Similarly, RWU is likely rather heterogeneous when considering the moisture changes in the rhizosphere as a 3D space. Hence an estimate based on one profile has clear limitations, too. We can follow the argumentation that the tree trunk is at least some sort of gauge where all water must pass. However, the SF processing involves quite some steps where the resulting fluxes can scale considerably.

**3 Bibliography**

Nguyen, H. H., Jeong, J., and Choi, M.: Extension of cosmic-ray neutron probe measurement depth for improving field scale root-zone soilmoisture estimation by coupling with representative in-situ sensors, Journal of Hydrology, 571, 679–696, https://linkinghub.elsevier.com/10retrieve/pii/S0022169419301751, 2019.

---

## Author Comment (AC2) · 14 May 2020

**1  General reply to all referees**

We sincerely thank Jesse Nippert, Jia Hu and Leander Anderegg for their intense study of our manuscript and their constructive feedback. We clearly understand that we have to simplify some of the dense writing and figures to convey our findings more clearly. The referees made several detailed suggestions for this, along which we will organise the revisions. We will self-critically check for simplifications of jargon and clarity in our arguments.

With respect to the observed process dynamics of measured and inferred variables,

we will carefully revise the manuscript towards i) a more detailed description of the observed results, ii) more coherent argumentation lines, and iii) the limits of the presented approach. We will put specific attention to a) the conversion of sap flow velocity and rhizosphere water withdrawal to flux rates, b) the assessment of the coherence of the diurnal signal with the assumed step shape, and c) the reference to inferred matric potential. At a meta-level, we have to make sure not to overstretch the data set at hand which is basically a first reference. We hope that many more researchers will employ, test and evaluate the proposed approach to estimate RWU, which together will form a more comprehensive picture of the complementary information in RWU, SF and ET.

**2 Specific replies to the review by Jia Hu**

The referee's comments are given in *italics* with our answers in regular font style.

*In this study, the authors examined root water uptake in beech trees, along a soil moisture profile in two sites with different moisture conditions. There were some interesting findings. First, they found differences in depth of water taken up by the roots between the two sites (sandy soils use deeper water). This was particularly evident during the period of maximum transpiration rates. They also found that while RWU could be estimated from changes in soil moisture, there were also instances where transpiration was occurring, but RWU was not measured. In a comparison between the sand versus slate sites, although SF was similar throughout the year, RWU was quite different.*

Thank you very much for this summary in which we see our study well-understood.

*One of the main ways in which this manuscript can improve is to clearly discuss the*

*reasons for comparing RWU and SF. The authors state that the aim of the study "is to evaluate the potential and limitations of the diurnal decrease of rhizosphere soil moisture measurements as an estimate for RWU in ecohydrological field studies." They make the point that RWU is an under measured observation, and that other proxies, such as changes in soil water content, are used to infer RWU. Meanwhile, sap flow sensors measure transpiration rates in trees, but because of stored water within trees, sap flow does not measure RWU uptake either. So linking RWU and SF (as mentioned in hypothesis 2) seems to be an important link. However, what wasn't clear for me is that if ET is an important metric to quantify for ecohydrological studies, what does RWU measurements provide that SF measurements don't? In other words, what additional processes related to ET do RWU measurements elucidate? I think this discussion could be enhanced more in the introduction. For example, in lines 28, the authors state, "Furthermore, spatially distributed monitoring of both RWU and soil moisture and SF could help to elucidate differences between the influence of the geological and pedological settings on water supply to transpiration and the influence of the plants themselves and their adaptations in root systems, dynamic sourcing of water and transpiration efficiency." Does this suggest that RWU influences the "geological and pedological settings on water supply to transpiration" while SF measurements assess the "influence of the plants themselves and their adaptation in roots systems...?" But SF also influences RWU, so shouldn't SF and RWU be considered in a framework that acknowledges that they influence each other?*

Thank you very much for raising our attention to this point. We see SF and RWU as communicating pairs of a common process, however, with slightly different foci. The method we propose to infer RWU from soil moisture measurements can help to assess the influence of (abiotic) site characteristics on the water availability for the tree. Additionally, assessing the dynamics of RWU from different depths also provides information on the hydrological conditions and processes within the rooting zone. In contrast, SF is mostly used as proxy for actual tree transpiration (with some uncertainty regarding tree water storage and assumptions during the calculations of sap flux), and

is also influenced by adaptations of the tree to the local site conditions. We therefore suggest to measure both SF and RWU for a better understanding of the water transport through trees. We will clarify this notion in the revisions.

*Figure 3. Why are estimates of Dq positive if the soil moisture decrease throughout the three day period? In page 6, line 22, does "change in soil moisture" refer to Dq? If so, again, why is Dq positive? The positive values of Dq during the daytime is confusing because in Figure 4B, Dq during the daytime hours is shown as negative. In Equation 1, the authors also state that a check to evaluate the data is that " day slope of soil moisture is negative (decline in soil moisture during the day)..."*

We agree that it is confusing that we define the change in soil moisture negatively in Fig. 3 but regularly in Fig. 4 and the calculation. We will clarify this in the revisions.

*Page 7, line 1. Do the bolded a) and b) here refer to a subset of "soil moisture (b)" from page 6, line 15? If so, I would change "a) and b)" to "i) and ii)" as to not confuse the reader.*

Thank you for the suggestion. We will change this as proposed.

*Page 7, line 4 and 5. No need to say "no STRONG decline in soil moisture" or "no TOO STRONG increase in soil moisture" since STRONG or TOO STRONG are quite subjective. I think that saying "no decline in soil moisture" or "no increase in soil moisture" followed by the rates of increase or decrease is sufficient.*

Thank you. We will change this as proposed.

*Page 7, line 30. Why was the assumption made that measured sap flow originates in the soil moisture decrease? Could there be any storage of water in the trunks (i.e.*

*might lags between RWU and SF exist)?*

We do expect some lags between RWU and SF due to water storage in the tree, that is why we have highlighted this (rather blunt) assumption. It appears difficult to quantify such a storage effect without further data (i.e. ET and more references of SF). However, with a temporal aggregation to daily steps we see a relatively high correlation between RWU and SF. Thus we do not expect the lag effect of water storage in the trunk to be very pronounced at this resolution.

*Page 7, line 31. "This is done by linear regression of daily sap flow to the sum of RWU over the soil profile with assumed zero intercept." Is the assumption again here that water from the different soil layers instantaneously feeds into the transpirational stream - in other words, there is no lag in when water is taken up by the roots and then transported to the trunk of the tree?*

It is generally correct that we neglect an intercept within the tree by applying a regression. However, since we sample the recorded data to daily aggregates, differences between the fluxes with shorter temporal footprint should cancel out (i.e. the lag between sap flow and RWU in Fig. 3). Hence "instantaneous" connection is not assumed. Nevertheless, we find strong differences in RWU and SF (Fig. 8), which might hint to water storage dynamics within the tree. However, we cannot assume to have sampled all sources of RWU with the soil moisture profile. Especially at the slate site it is very likely that roots can source water from local subsurface pools or films in the gravelly subsoil. We will clarify these points in the revised version of the manuscript.

*Page 8, Line 1. "The resulting factor is the mean reference area required to supply to observed sap flow." Is the 'factor' mentioned here the area or the volume? If RWU is summed across the different soil depths in which soil moisture is measured, how is the resulting factor estimated as area and not volume?*

As stated in the mentioned subsection a proper comparison of SF and RWU requires them to be defined as fluxes. This means that we have to refer to a cross-sectional area of active xylem for SF and a reference rhizosphere volume for the observed change in soil moisture attributed to RWU. Here the height of each volume increment is given by the integration length of the soil moisture profile probe, which is 0.2 m. Without knowledge about the actual root distribution we simply assumed a cylindrical rhizosphere. The "factor" is hence the projected area of this cylinder which can be expressed as radius for a plausibility check (see legend in Fig. 7). Since the RWU is defined in mm/day (a volume normalised by the area) the factor has to be an area to derive the volume flux. We will reconsider how this step can be clarified.

*Page 10, Line 2. "In later summer, the RWU signal ceases although the sap flow signal continues at lower rates." In Figure 6, I don't see when this occurs across the entire instrument period.*

The visual comparison of sap flow (L/day) and RWU (mm/day) dynamics has its drawbacks. This is why we opted to extend the analysis with the estimate for fluxes instead of the direct signals. However, it is not clear how much the assumptions to derive the volume fluxes will blur the actual signal in the observations. We agree that this statement can be seen as subjective. As Fig. 6 is subject to revisions, we will revise the interpretation accordingly and opt to refer to the following analyses in Fig. 8 instead.

*Page 10, Line 32. "With a working-hypothesis of a closed water balance...the linear regression also results ....at the sandy site the cylinder would have a radius of 4.2m...slate site one would estimate a radius of 5.5m." I may have missed this, but how did you reach these readius values? Where is the linear regression model reported? I see that there are radius values reported in Figure 7, but how were these calculated?*

Please see above (comment to P8L1). We will clarify this step in the revised manuscript

accordingly.

*Page 12, Line 2. "However, the high initial correlation drops in July. At the sand site, this marks the shift to RWU ranging below SF. At the slate site, no such transition is apparent." In Figure 8, when the spearman correlation drops, the precedes when RWU drops below sap flow. There are also instances later in July when RWU is consistently below SF but the spearman correlation ratio does not change. What does this mean?*

The Spearman rank correlation exactly "punishes" the change in ranks. Frequent changes result in low correlation values (e.g. August at the slate site). When RWU is consistently below OR above SF the correlation can become rather high. Since this is not giving the full picture, we report the KGE as alternative measure of correlation which "punishes" deviation of the dynamics and the absolute values.

*Page 14. Line 9. I would recommend changing the work "ambivalent" to "mixed."*

Thank you. We see the awkward wording and will change it.

*Page 16, line 9. "What is the optimization function of the plant's RWU sourcing and SF variability?" What do the authors mean by this? Please explain.*

Gao et al. (2014) show that climate leads to an adaptation of the rhizosphere storage capacity. Saveyn et al. (2008) show how different SF can take place in the xylem under different weather conditions. We agree to your argument that RWU and SF have to be considered as interactive processes. Hence we expect the plants to adapt to climatic and site conditions. We expect that this adaptation is not a random process but some sort of optimisation.

*Page 16, line 12. Yes, wounding from sap flow sensors can indeed underestimate*

*sap flux velocity, and non-homogenous xylem depths can influence estimates of total transpiration rates, but it seems unlikely that these effects would be most noticeable during periods when both sap flux and RWU begin to decline. The authors allude to other factors in the previous paragraph (e.g. stem storage, leaf level transpiration) that offer more likely explanations for why correlations between RWU and sap flux correlations decrease as the soils dry out.*

Thank you for your evaluation of these influencing factors. We will carefully check that affecting factors are presented in a balanced manner.

**3 Bibliography**

Gao, H., Hrachowitz, M., Schymanski, S. J., Fenicia, F., Sriwongsitanon, N., and Savenije, H. H. G.: Climate controls howecosystems size the root zone storage capacity at catchment scale, Geophysical Research Letters, 41, 2014GL061 668–7923,https://doi.org/10.1002/2014GL061668, 2014.

Saveyn, A., Steppe, K., and Lemeur, R.: Spatial variability of xylem sap flow in mature beech (Fagus sylvatica) and its diurnal dynamics in re-5lation to microclimate, Botany, 86, 1440–1448, https://doi.org/10.1139/B08-112, 2008.
* * *

---

## Author Comment (AC3) · 14 May 2020

**1  General reply to all referees**

We sincerely thank Jesse Nippert, Jia Hu and Leander Anderegg for their intense study of our manuscript and their constructive feedback. We clearly understand that we have to simplify some of the dense writing and figures to convey our findings more clearly. The referees made several detailed suggestions for this, along which we will organise the revisions. We will self-critically check for simplifications of jargon and clarity in our arguments.

With respect to the observed process dynamics of measured and inferred variables,

we will carefully revise the manuscript towards i) a more detailed description of the observed results, ii) more coherent argumentation lines, and iii) the limits of the presented approach. We will put specific attention to a) the conversion of sap flow velocity and rhizosphere water withdrawal to flux rates, b) the assessment of the coherence of the diurnal signal with the assumed step shape, and c) the reference to inferred matric potential. At a meta-level, we have to make sure not to overstretch the data set at hand which is basically a first reference. We hope that many more researchers will employ, test and evaluate the proposed approach to estimate RWU, which together will form a more comprehensive picture of the complementary information in RWU, SF and ET.

**2 Specific replies to the review by Leander Anderegg**

The referee's comments are given in *italics* with our answers in regular font style.

*In this manuscript, the authors pair soil moisture measurements that have high spatial and temporal resolution with tree sapflow measurements at two different sites to test whether such soil moisture measurements can be used to estimate daily transpiration and identify depths of root water uptake (RWU). They find promising similarities between sap flow and estimated RWU during a fairly wet period at their site with sandy soil, but worse correlations at a site with more heterogeneous soil characteristics and a time series that extended into a drier period. They also found interesting evidence for differences in the depth of RWU at the two study sites, though this is somewhat deemphasized in the text. While the estimated daily RWU uptake appears promising in some regards, they also found a confusing lack of relationship between RWU and soil matric potential calculated from soil moisture release curves and soil water content. All told, these results suggest that the method is promising but still has some kinks to be*

*worked out, some of the largest probably relate to spatial heterogeneity at large scales (lateral variation over meters) and fine scales (inability to infer matric potential from soil moisture release curves on nearby soil samples.*

*This is an interesting manuscript that presents a promising approach to estimating transpiration and RWU at high temporal scale. However, my three main concerns are: 1) The writing and figures are extremely dense and sometimes confusing/contradictory. I had to read the Results at least two times, and often had to parse out individual sentences multiple times before I could begin to follow their meaning. Some of this could be due to a difference in fields (hydrology vs the plant ecophysiology terminology that I am more familiar with), but I would recommend a considerable expansion of the Results to explain the more complicated an nuanced findings and make this interpretable by a broad audience. I have given multiple suggestions below in the 'Specific Comments' section, but would general recommend a careful edit and clarification of the most complicated sentences in the Results. I also would recommend simplifying some of the figures by breaking out aspects into multiple panels rather than layering on 4-5 different sources of information that I found almost impossible to interpret simultaneously. In particular, Fig 6 and 12 are nearly impenetrable (and Fig 11 is also quite dense).*

Thank you for your intense study of our manuscript and taking the challenge to dig out our messages so well. We gratefully receive your suggestions to clarify and simplify the manuscript including some of the figures for better understanding.

*2) The introduction oversells the novelty of monitoring soil moisture to estimate RWU dynamics and transpiration. True, the ability to monitor soil moisture with high enough precision to assess daily RWU is fairly novel and new, but people have been measuring soil moisture to estimate depths of RWU and understand transpiration budgets for decades! In fact, I would argue that gravimetric or volumentric soil moisture measurements are the original method for estimating transpiration (e.g. just to name a couple that come up with a quick google search: Denmead & Shaw (1962) "Availability*

*of soil water to plants as affected by soil moisture content and meteorological conditions" Agronomy Journal; Novak (1987) "Estimation of soil-water extraction patterns by roots" Agricultural Water Management). Thus, I think it is important in the Introduction to stress that it is the precision of these measurements (allowing high temporal and spatial estimation of RWU) that is interesting, not the method and theory itself.*

We generally agree to this point and will seek for a more balanced presentation. Despite the clear reference of RWU to soil water content our aim is to highlight the capability of this easily available technique for such analyses - given the level of precision and spatial coherence. We will revise the introduction accordingly.

*3) I think the authors do a good job honestly discussing where their approach did not perform well, but I would both urge them to focus and structure the discussion around a coherent argument for what the key processes and attributes are that screw up these measurements (e.g. what are the 4 biggest problems, list them out, and show us how you concluded that these are what is causing the method to fail at the Slate site and in dry soils).*

Thank you for acknowledging our efforts. As often, one quickly arrives at "it depends" when distilling such 4 biggest problems. Actually, the system is underdetermined given just some SF and soil moisture sensors to rigorously conclude such a fixed list. However, we will revise the manuscript to convey these arguments in a stringent and clear manner.

*I would also urge the authors to reconsider the framing and discussion around their 'Hypothesis 3'. It is currently framed as an open question whether tension gradients drive variation in root water uptake. And then Figure 11 is presented as evidence that this may not be the case. I think this is a misrepresentation of both where the field is at and what the confusing findings of Figure 11 represent. Plants can alter RWU via*

*changes to root properties (changing aquaporin expression to alter root permeability) and root distribuitons, but they cannot physically fight potential gradients as the authors seem to suggest with Fig 11 and in the Discussion. Plants can ONLY extract and move water by moving it down a potential gradient, and there is no physical way the plant can be extracting and transpiring water from soils with a matric potential 10s-100s of MPa below 'permanent wilting point' (âĽij1.5 MPa, or 4.2 (log10(hPa)). The general dogma (assuming +/- equivalent root resistances throughout the soil profile) that water uptake by roots should be proportional to the pressure difference and the root surface area/biomass should be used as a final test for the reliability of this method to estimate RWU, rather than using the data to test the dogma. In this case, I think it is painfully obvious that we have essentially no reliable way to convert water content to matric potential at the spatial and temporal scales that are relevant to these transpiration estimates. In fact, we're SO BAD at it, that it would appear that the Slate trees are extracting water from soil with a matric potential of Ân -10 MPa (when leaf water potential, the ultimate pressure differential driving water movement, is almost certinaly > -2 MPa). That tells me that there's a problem with the method, not the theory. However, recognizing this allows you to say something interesting about why we can't back calculate matric potential from these measurements (spatial heterogeneity in soil properties? Problems with rock fractions? Rock fractures that don't behave like soil samples used for dehydration curves?).*

Yes. We fully agree to this notion and clearly see the implications for our hypothesis 3. In a different study we exactly work out this point that it has been somewhat "forgotten" to intensify research about how to measure matric potentials but that both variables are essential to define soil water state dynamics because single retention functions render inconclusive – as in our example here. Likewise, measuring matric potential in general and in heterogeneous soils (like at the slate site) especially is challenging and could not be done in a similar manner as the soil moisture profile in this study. We will follow your advice and will reconsider if and how we should include a reference to matric potential inferred from soil moisture measurements in our study.

*Specific comments: Pg 6 L11-14: Please explain a little more what you mean by 'NSE is a measure which is very sensitive to deviations from shape features" (perhaps you could add a day that does not pass this cuttof to Fig 4 to illustrate?), what cutoff of NSE you used, and how you arrived at that cutoff.*

Thank you for the suggestion. We will add further reference candidates to Fig 4B to exemplify the differences and evaluation criterion.

*Pg 7 L8-12 and pg10 L30 and Fig 7: I am very confused about what 'corrected' means. In the Methods, I interpreted 'Corrected' to mean RWU extrapolated from the linear regression through the nightly data (magenta line in Fig 4a). But in Fig 7 the 'not corrected' values (blue points) are higher than the 'corrected' values (colored points), which tells me I'm getting confused somewhere. Please clarify this in Fig 4, and Fig 7 and the associated Results text (pg 10 L30).*

The difference between "corrected" and "not corrected" is if the slope of the nocturnal phase is extrapolated to form the reference or if simply the difference between the reference time stamps is calculated. Given our criteria for the step shape, the nocturnal phases can have slight negative slopes, which would lead to the observed situation that the not-corrected RWU is higher. This hints to phases when some diffusive percolation is taking place. With more description of the results in the revised manuscript, we will also clarify this.

*Pg 7 L20-29: This paragraph about turning sap velocity into sap flux is very confusing. I did not understand it until I scrutinized Fig 5. Please rewrite/clarify. Also, in the Fig 5 legend/caption it is worth noting that the "5mm, 18mm, and 30mm" are depths from the outside of the tree (or inside of inner bark? Not sure which).*

Again thank you for diving deep. Before installing the sap flow sensors, the bark is removed, so the depths are measured approximately from the cambium. We will restructure the presentation of this step to make it easier to follow.

*Figure 6: I had a very difficult time extracting the desired inferences from this figure. The shading (which varies per site, over time, and in different soil layers) is almost impossible to see and interpret (not to mention some of the colors become colors used for other soil depths when shaded) yet are referenced multiple times in the text. Also, the stacked bar plots make it almost impossible for me to interpret which depths are providing RWU, mostly I just take away total bar height. I would recommend 1) breaking out the information about how well the RWU estimation likely worked into another panel or method other than shading (filled versus unfilled bars/symbols, perhaps?). and 2) either finding a more holistic way of showing depth information (e.g. coloring whole bars by the weighted average depth of RWU) or just making a different panel that showed line graphs of uptake by depth through time. In fact, I would potentially advocate for breaking out the depth of extraction information into a new figure altogether.*

We take this point and will rethink this figure. First of all, the shading will be dropped and given in a further figure. Using line plots instead does not help comprehension as this was our first try on this. We will reconsider how to make this legible.

*Pg 12 L14-15: I don't quite understand what data are being compared in this sentence "Comparing RWU correlation between the two sites, applying the nocturnal correction improves Spearman rho form 0.42 to 0.52. KGE remains almost the same with 0.27 increasing to 0.3." All data from both sites (if so, why is this a useful comparison)? Or somehow site-level averages?*

Obviously, we have to clarify this in the revisions, thank you for pointing it out. In the analyses before we have used the "corrected" data. Here we confirm that our findings about low correlation between RWU of the two sites remain valid with either approach. We see that the confusion might originate from this rather technical detail concluding

the site and process comparison subsection. This will be transferred to the discussion section.

*Section 4.3 – I think this section is very cool, but I understood very little of the text. What does "a diffuse redistribution into the surrounding soil aggregates" mean and why can it be "seen as parallel declines. . .in the different depth layers"? Please explain more what "flashy transport through the macroporous soils and fill-and-spill mechanisms of subsurface pools" means, and much more importantly how this analysis influences our interpretation of the method for assessing RWU in this site. Clearly you learned something interesting and highly relevant (possibly that helps us interpret Fig 11?), but I do not understand what it is based on the current text. For instance, I have no idea what these sentences mean and how they relate to Fig 10b "Here, roots are likely to grow along joints and fractures, where event-water can be stored with little effect on the bulk soil moisture. As such, the measurements might miss parts of the active rhizosphere."*

The hydrological jargon might be especially dense here - also because we expect quite some idea about the sites from the reader. We will explain in more detail in the revised version and reconsider if this might become part of the site description (again).

*Section 4.4 – See above comments about interpretation and framing of these results. Also, the current Figure 11 is nearly impossible for me to interpret. I would recommend displaying SOME aspect of this information in multiple panels. (e.g. maybe splitting the soil columns up into three depths and displaying them as separate panels so you can color by SF). Also, the units/label on the x axis of this figure is confusing to me. And honestly, after reading the text of this section 4 times, I still don't have any idea what it means. I can't even decipher it enough to make suggestions on how to clarify it. I don't know what the referenced 'reactions' are and how I'm supposed to assess them in the figure. Moreover, I do not at all see the 'correlation of matric potential and depth' that*

*supposedly exists in slate site.*

We will re-evaluate Fig. 11. As you point out, we also need to rework the whole argument showing that it is not the plants sourcing at high flux rates against physiologically impossible tensions but the conversion of soil moisture into matric potentials, which does not represent the state around the roots. We will take care of this in the revision.

*Pg 14 L14: This sentence "At the same time, we pointed out considerable limitations to the approach with respect to soil water state (no detectable signal during low moisture periods) and soil properties (high variability in heterogeneous soil profiles)" is the most interesting sentence of the discussion to me, but comes out of no where and needs much more explanation. In order for me to follow your train of thought, I require much more explanation. . .*

Thank you for highlighting this lack of reference. We will build the links for clarification.

*Figure 12 and associated text of Section 5.1: I had an extremely hard time interpreting this figure. Please 1) remove the red bars for total extraction to new panels (two axes y-axes with different interpretations is much more than my brain can handle). 2) Explain what the NSC cutoffs indicate, and what the larger blue bar for 'all detected' is and why the inset bars for different detection thresholds do not sum to it 3) Put panel A and B on the same axis (e.g. 0%-90%) and switch the big numbers to be % of days and little numbers to be of days. Also, how does Fig 12 show "The RWU derivation function appears to perform very well in general and can be used to evaluate a broad range of diurnal changes in soil moisture (Fig. 12)." (L1-2). Moreover, this sentence doesn't really make sense to me "Unlike the first impression in Fig. 6, the proportion of steps with higher uncertainty about the actual fit of the shape with the assumptions is higher in the slate site data, which is in line with the lower overall RWU detection there." Could you explain what you mean by "higher uncertainty about the actual fit"? Also,*

*how "uncertainty" and rate of "overall detection" differ? Throughout this section, please be much more explicit about the site, times, and layers you are referring to when, for example, you write "Under somewhat ideal conditions with soil moisture sensors and roots in good contact with a rather homogeneous soil matrix and sufficient soil water availability, the diurnal steps are identified and evaluated with great confidence." Finally, this feels like it should be in the Results, perhaps even near Figure 1, rather than in the Discussion.Pg 15- L5: I think it's worth explicityly mentioning the take-away from Figure C1: that flux amount is unrelated to how well the step function fits the daily soil moisture pattern.*

Again, a big thanks for working out the twists which have made their way into our argumentation. Emphasising on the RWU estimation in the revised manuscript will make room to clarify on this as one of the main results. Thank you also for the suggestions on how to make Fig. 12 easier to understand. We will consider them for the revision.

*Pg 16-L25-35: See my comments about Hypothesis 3 and Figure 11. Also, the sentences at L28 ("At the sandy site. . .") seem confusing and almost self contradictory to me.*

We expected to find some evidence for a preference of cheap water (large fluxes at low tensions) by the roots. The sandy site somewhat depicts that. However the periods of implausibly high tensions have to be reconsidered. At the slate site, gravel content (which increases with depth) is a likely explanation for the "strange" picture. When we correct the available pore space with gravel content the whole distribution should shift to the left and make more sense. We will check both avenues: i) if the general argument around hypothesis 3 is fruitful and ii) how to clarify on Fig. 11 and the respective result description.

---

## Author Response (AR1)

**Revision of "Estimates of tree root water uptake from soil moisture profile dynamics" by Conrad Jackisch et al.**

We again sincerely thank Jesse Nippert, Jia Hu, Leander Anderegg and Chris Still for their constructive comments to our manuscript. We have revised it to convey our findings more clearly.

The revisions are aligned with the main suggestions of the reviewers i) to simplify the presentation, ii) to make the lines of arguments more coherent and iii) to disentangle some of the complicated plots.

Following the suggestion of Leander Anderegg, we have omitted the aspect of sourcing of RWU with reference to matric potential inferred from measured soil moisture and water retention functions. The obvious disagreement of an apparent high tension with high root water uptake (RWU) is an issue to look into. However, it does not contribute to the focus of this study. We have moved the respective information into the Appendix.

Especially Jesse Nippert pointed to the dense information in our figures (and writing). We have revised the complex figures and recompiled new versions, which may still not appear completely intuitive at first sight. With more detailed figure captions and clearer alignment with the methods and results, we hope they are now easier to understand and digest. Specifically, we have now condensed the different steps of calculating RWU and sap flow (SF) in one figure each (Fig. 4 and Fig. 5), in line with the description in the text. The revised Fig. 6 still presents the time series of daily SF and RWU of the two sites. With clear reference of the notations to Fig. 4 C and Fig 5 C, we hope the information is easier to grasp. We also included extra panels for the NSE evaluation and a reference to plant available water content which should help to understand these results.

Because the different methodological steps to distil a comparable flux rate of RWU and SF from respective dynamics of soil moisture and sap velocity caused some confusion, we took special care to lay out these steps more clearly. We explain the methods step by step and describe the corresponding results in more detail.

**Specific revisions in reply to the referees:**

The referees' comments are given in *italics* with our original replies in in regular font style. For the most part we followed our initial replies to the comments. In the following we only describe our final changes to the manuscript. Please consider the original replies of the discussion phase if you would like to compare the two. Only comments which required revisions are listed.

**Short comment by Jesse Nippert**

I have few major concerns or suggestion revisions for the authors to address. My most noteworthy suggestion for the authors is to simply / clarify this manuscript whenever possible. This is an extraordinarily detailed and jargon-rich manuscript that needs to be simplified. I'm a scientist that studies plant-water use (so in theory, this manuscript is directly within my field of study) and yet I found myself reading and re-reading sections to try to understand what was done, what the data means, and how the authors derived their conclusions. For many of the figures, I was never able to gain a full appreciation of what was conveyed (what exactly am I seeing in Figures 11 and 12?) and the legends were often non-descriptive (see Fig. 4 for an example). Thus, I encourage the authors to reduce the jargon, better explain the development of the RWU calculation, and simplify the figures whenever possible. The novelty and creativity of this manuscript is in the top 95%, but the clarity and delivery of the information is in the 50-60th percentile. I'm fully confident the authors can address this issue.

We have worked through our manuscript and simplified the general structure by moving the section about soil moisture dynamics to the site description and by omitting the aspects of soil water retention. By doing so, we hope to reduce much of the cluttered arguments. Moreover, we have extended the descriptions in the figure captions, removed jargon where possible or explained more clearly elsewhere.

Other comments: The Introduction is generally sound, but there are a few items of concern. On page 1 line 17, the authors need to remove 'even over grasslands'. Grasslands are not definitely simpler ecosystems that forests, so please do not refer to them as such.

We agree to the point that grasslands are not simpler ecosystems than forests and that this wording was not well chosen. We have removed the statement.

On line 18, please address the 'optimise their water transport to respiration'. The optimality theory has been challenged many times since 2009, and in fact doesn't appear to be valid. In addition, why would water transport be optimized for respiration? I believe you meant assimilation. Regardless, please update.

You are right, we refer to assimilation here. To make that clear we omit the debate about optimality in the revised manuscript and rephrased the respective passage.

Examples of RWU assessments in the literature are provided on page 2. While I have never read an approach as detailed as the one presented here, transpiration dynamics have been assessed by comparing sap flux with changes in soil moisture dynamics previously. Please check out Holdo and Nippert 2015 Ecology (https://doi.org/10.1890/14-1986.1) for a study comparing transpiration dynamics using sap flux, soil moisture, isotopes, and changes in canopy temperature among coexisting trees and grasses.

Thank you for pointing to this study. After considering it, we do not think it reflects what we want to say here. In the revised manuscript we have inserted statements clarifying the novelty of the approach to reside in the automated derivation of RWU from soil moisture declines in a continuous depth profile – but not the relation of transpiration and soil water status in general. We point to some of the studies which use, for example, isotope methods to learn more about the detailed path of water from different soil depths into trees.

On page 5/line 7, the authors note that sap flux was monitored in 4 trees near the TDR probes at each site. Do subsequent sap flux data (Fig. 3, 5, 6, etc) represent a single tree, or an interpolation across all 4 trees per site?

Yes, the respective data represents a single tree. Originally, we intended to use the tree where we directly measure soil moisture in the rhizosphere. At the slate site we had to change this because of a sensor failure. We justify switching to a different tree with the high correlation of sap velocities among the trees (as shown in Appendix A). We describe the selection of the tree in more detail now in section 2.3 and Appendix A.

Section 3 describes Fig. 3, and the derivation of RWU based on soil moisture step change. How was this 3 day period selected? Can I assume it was the best 3-day period where a step change was observed during this summer? Does the lower correlation in RWU time series (and during drier times) reflect derivation of the metric during an ideal period, which then loses predictive power during mean summer soil moisture periods?

This exemplary period in Fig. 3 has been chosen arbitrarily as it combines clear sky conditions (day 1), clear sky with an intermediate shading (day 2) and a fair weather with radiation noise by smaller cumulus clouds (day 3). The step changes have been observed at many more days. We have introduced a set of simple criteria to define a general step.

Afterwards, we evaluate if the soil moisture declines follow an idealised step shape (not an ideal time period) which we would attribute to mainly root water uptake, as opposed to strong deviations from the step pointing towards an interplay of various processes. To this end we calculate the NSE between the data and an idealised step for each day. The number of days meeting our simple criteria for a general step are summarised in Fig. 6 and 7, along with the additional overlay of the NSE classes from the evaluation step.

Our proposed RWU calculation approach indeed loses predictive power when the changes in soil moisture become relatively small and hence our assumption of the diurnal steps is no longer met. This changes with time and depth layer. Roughly summarised: We find steps in the data with an NSE persistently >0.5 between May and July at the sand site. Also the slate site reports higher NSE values in this period. However, there the level of determination is generally lower. These temporal dynamics are now given as separate panels in Fig. 6.

The NSE evaluation we undertake here is a fundamentally different step from the correlation analysis between estimates of RWU and SF based on KGE (Fig. 9). We have reworked the presentation of methods and results and hope to convey this information now.

When linking RWU from TDR data to transpiration dynamics from an individual tree, how did you account for water use by the understory vegetative community? Unless there were no grasses, forbs, saplings, etc. – wouldn't these species be using water from the same soil depths which would then complicate predictions of individual tree water-use using changes in soil moisture? The methods section does not describe the understory.

We added a sentence in the discussion stating that there was barely any understory vegetation so this effect should be negligible.

With respect to evaporation we expect an effect on the top 0.2 m. However, since the signal from this layer is only rarely evaluated as RWU we are quite confident to be correct here. From a discussion point for the proposed approach, we agree that in many applications with understory vegetation the soil moisture dynamics cannot differentiate between different plants.

On the sandy site, how did you account for capillary action within the soil profile and subsequent evaporation? Or do you have some information (which I may have missed) on evaporation rates, and the depths of the evaporative fronts from these soils?

We do not have any reliable evaporation reference. There is likely some evaporation from the top soil layer which we do not assess but might see in our graphs. We have added a paragraph about this in the discussion section 5.1

Fig. 6 is an impressive behemoth. One item of interest to me is that the rainfall history does not appear to have any temporal synchrony with RWU or sap flux. Is this true? And if so, why not?

We have reformatted the former Fig. 6 and removed the shading of the columns. We now present the NSE values of the step shape as separate panels. Moreover, we greatly extended the description in the caption and in the results section 4.1 and hope the figure is understandable now.

With regard to precipitation, we mainly see its effect on the radiation input and a resulting decreased SF. We also see some increase of SF after rainfall in autumn, which might have been more available water after a dry summer. We added an indicator for available water to the graph and describe the possible influence of precipitation in the results section 4.1 now.

In Fig. 7, how does these predictions of water use in mm/day compare to estimates of Beech from similar locations within the literature. It would be nice to know if these predictions fall in the range reported elsewhere.

We have added literature values for general reference. However, the spectrum of plausible flux rates is rather broad. Given the dependency on site and tree characteristics, we still are confident that our estimates are realistic but cannot provide any more detailed reference.

Page 14, line 9 – I don't think 'ambivalent' is the correct term. An ambivalent picture suggests this was inconclusive research. You concluded many things, and illustrate a path forward for using soil moisture to infer plant water use.

Thank you for pointing to this improper wording. We have now termed it 'nuanced'.

Page 17, line 5. How do you know it's a minor effect? HR can be quite substantial in many ecosystems.

We agree that hydraulic redistribution in the rhizosphere can be substantial and that especially at the sandy site we might miss important factors by neglecting it. We rephrased this and now discuss the general possibility of HR in section 5.3.

Page 17, line 14 – How would cosmic ray measurements be appropriate here? I was under the impression that cosmic ray data pertains to the top 10 cm of the soil only.

We refer to attempts of combining cosmic ray measurements with in-situ soil moisture measurements to overcome the point information towards a better spatial representation like the authors in Nguyen et al. (2019) propose. However, we agree that the link might be rather far-fetched. We removed it and adjusted the outlook accodingly.

In the Discussion section, the authors note that RWU and sap flux were not inter- changeable, but were complementary. The language infers that RWU may be more de- sirable than sap flux for estimating transpiration under certain conditions. This seems a bit misleading to me. Sap flux directly measures a physiological process that relates to canopy transpiration, while RWU is an inferred metric. Under what conditions would the data from RWU be preferred over sap flux? Under what conditions does RWU outperform sap flux for predictions of transpiration?

We do not think that RWU is more informative than sap flow. Certainly both have their merits and drawbacks. We aim to suggest to refer to both means as complementary gauges of a highly interlinked process. Unfortunately, SF does not directly measure the physiological process but a flow velocity over a more or less difficult to guess cross-section. Similarly, RWU is likely rather heterogeneous when considering the moisture changes in the rhizosphere as a 3D space. Hence an estimate based on one profile has clear limitations, too. We can follow the argumentation that the tree trunk is at least some sort of gauge where all water must pass. However, the SF processing from the initial interpretation of the heat pulse advection to flow velocity and subsequent attribution of an estimated cross sectional area of active sap wood involves a series of calculations and assumptions, and the resulting fluxes can vary considerably. We revised the discussion to clarify that we consider both RWU and SF measurements equally justified, as complementary measures.

**Referee comment by Jia Hu**

One of the main ways in which this manuscript can improve is to clearly discuss the reasons for comparing RWU and SF. The authors state that the aim of the study "is to evaluate the potential and limitations of the diurnal decrease of rhizosphere soil moisture measurements as an estimate for RWU in ecohydrological field studies." They make the point that RWU is an under measured observation, and that other proxies, such as changes in soil water content, are used to infer RWU. Meanwhile, sap flow sensors measure transpiration rates in trees, but because of stored water within trees, sap flow does not measure RWU uptake either. So linking RWU and SF (as mentioned in hypothesis 2) seems to be an important link. However, what wasn't clear for me is that if ET is an important metric to quantify for ecohydrological studies, what does RWU measurements provide that SF measurements don't? In other words, what additional processes related to ET do RWU measurements elucidate? I think this discussion could be enhanced more in the introduction. For example, in lines 28, the authors state, "Furthermore, spatially distributed monitoring of both RWU and soil moisture and SF could help to elucidate differences between the influence of the geological and pedological settings on water supply to transpiration and the influence of the plants themselves and their adaptations in root systems, dynamic sourcing of water and transpiration efficiency." Does this suggest that RWU influences the "geological and pedological settings on water supply to transpiration" while SF measurements assess the "influence of the plants themselves and their adaptation in roots systems...?" But SF also influences RWU, so shouldn't SF and RWU be considered in a framework that acknowledges that they influence each other?

Thank you very much for raising our attention to this point. We see SF and RWU as interrelated elements of the transpiration process, however, with slightly different foci. The method we propose to infer RWU from soil moisture measurements can help to assess the influence of (abiotic) site characteristics on the water availability for the tree. Additionally, assessing the dynamics of RWU from different depths also provides information on the hydrological conditions and processes within the rooting zone. In contrast, SF is mostly used as proxy for actual tree transpiration (with some uncertainty regarding tree water storage and assumptions during the calculations of sap flux), and is also influenced by adaptations of the tree to the local site conditions. We therefore suggest to measure both SF and RWU for a better understanding of the water transport through trees. We have clarified this in the discussion of the revised manuscript.

Figure 3. Why are estimates of Dq positive if the soil moisture decrease throughout the three day period? In page 6, line 22, does "change in soil moisture" refer to Dq? If so, again, why is Dq positive? The positive values of Dq during the

daytime is confusing because in Figure 4B, Dq during the daytime hours is shown as negative. In Equation 1, the authors also state that a check to evaluate the data is that " day slope of soil moisture is negative (decline in soil moisture during the day)..."

We agree that it is confusing that we define the change in soil moisture negatively in Fig. 3 but regularly in Fig. 4 and the calculation. We hope to have resolved this confusion with a respective statement in the caption of Fig. 3.

Page 7, line 1. Do the bolded a) and b) here refer to a subset of "soil moisture (b)" from page 6, line 15? If so, I would change "a) and b)" to "i) and ii)" as to not confuse the reader.

We have changed this as proposed.

Page 7, line 4 and 5. No need to say "no STRONG decline in soil moisture" or "no TOO STRONG increase in soil moisture" since STRONG or TOO STRONG are quite subjective. I think that saying "no decline in soil moisture" or "no increase in soil moisture" followed by the rates of increase or decrease is sufficient.

We have rephrased the paragraph clarifying the idea and avoiding subjective claims.

Page 7, line 30. Why was the assumption made that measured sap flow originates in the soil moisture decrease? Could there be any storage of water in the trunks (i.e. might lags between RWU and SF exist)?

We do expect some lags between RWU and SF due to water storage in the tree, that is why we have highlighted this (rather blunt) assumption. It appears difficult to quantify such a storage effect without further data (i.e. ET and more references of SF). However, with a temporal aggregation to daily values we see a relatively high correlation between RWU and SF (Fig. 7). Thus, we do not expect the lag effect of water storage in the trunk to be very pronounced at the temporal resolution of one day. In other words, the sap flux of several litres per day appears to be a much stronger signal than the water storage dynamics within the tree trunk. We have now clarified the issue of storage in the tree in the introduction and specified our methodological assumption of a closed water balance over the day.

Page 7, line 31. "This is done by linear regression of daily sap flow to the sum of RWU over the soil profile with assumed zero intercept." Is the assumption again here that water from the different soil layers instantaneously feeds into the transpirational stream – in other words, there is no lag in when water is taken up by the roots and then transported to the trunk of the tree?

It is generally correct that we neglect an intercept within the tree by applying a regression. However, since we sample the recorded data to daily aggregates, differences between the fluxes with shorter temporal footprint should cancel out (i.e. the lag between sap flow and RWU in Fig. 3). Hence an "instantaneous" connection is not assumed. Nevertheless, we find strong differences in RWU and SF (Fig. 8), which might hint to water storage dynamics within the tree. However, we cannot assume to have sampled all sources of RWU with the soil moisture profile. Especially at the slate site it is very likely that roots can source water from local subsurface pools or films in the gravelly subsoil. We have largely rephrased the section for more clarity.

Page 8, Line 1. "The resulting factor is the mean reference area required to supply to observed sap flow." Is the 'factor' mentioned here the area or the volume? If RWU is summed across the different soil depths in which soil moisture is measured, how is the resulting factor estimated as area and not volume?

As stated in the mentioned subsection, a proper comparison of SF and RWU requires them to be defined as fluxes. This means that we have to refer to a cross-sectional area of active xylem for SF and a reference rhizosphere volume for the observed change in soil moisture attributed to RWU. Here the height of each volume increment is given by the integration length of the soil moisture profile probe, which is 0.2 m. Without knowledge about the actual root distribution we simply assumed a cylindrical rhizosphere. The "factor" is hence the projected area of this cylinder which can be expressed as radius for a plausibility check (see legend in Fig. 7). Since the RWU is defined in mm/day (a volume normalised by the area) the regression factor has to be the area to derive the volume flux. We have rephrased the methodological description of this regression-based derivation (section 3.4) and also clarified for which further analyses we use these assumed rhizosphere dimensions.

Page 10, Line 2. "In later summer, the RWU signal ceases although the sap flow signal continues at lower rates." In Figure 6, I don't see when this occurs across the entire instrument period.

The visual comparison of sap flow (L/day) and RWU (mm/day) dynamics has its drawbacks. This is why we opted to extend the analysis with the estimate for fluxes instead of the direct signals. However, it is not clear how much the assumptions to derive the volume fluxes will blur the actual signal in the observations. We agree that this statement can be seen as subjective.

Along the revisions and extension of the methods and result description, we have clarified the signal interpretation to avoid subjectivity. Moreover, we revised the respective figures, to make them easier to follow.

Page 10, Line 32. "With a working-hypothesis of a closed water balance...the linear regression also results ....at the sandy site the cylinder would have a radius of 4.2m...slate site one would estimate a radius of 5.5m." I may have missed this, but how did you reach these readius values? Where is the linear regression model reported? I see that there are radius values reported in Figure 7, but how were these calculated?

We have revised the presentation of the regression in the methods section 3.4.

Page 12, Line 2. "However, the high initial correlation drops in July. At the sand site, this marks the shift to RWU ranging below SF. At the slate site, no such transition is apparent." In Figure 8, when the spearman correlation drops, the precedes when RWU drops below sap flow. There are also instances later in July when RWU is consistently below SF but the spearman correlation ratio does not change. What does this mean?

The Spearman rank correlation "punishes" the change in ranks. Frequent changes result in low correlation values (e.g. August at the slate site). When RWU is consistently below OR above SF the correlation can become rather high. Since this is not giving the full picture, we report the KGE as alternative measure of correlation which "punishes" both the deviation of the dynamics and the absolute values. We extended the description of the results with special focus on the correlation measures and hope to be clearer now.

Page 14. Line 9. I would recommend changing the work "ambivalent" to "mixed."

Thank you. We changed the wording to 'nuanced'.

**Page 16, line 9. "What is the optimization function of the plant's RWU sourcing and SF variability?" What do the authors mean by this? Please explain.**

Gao et al. (2014) show that climate leads to an adaptation of the rhizosphere storage capacity. Saveyn et al. (2008) show how different SF can take place in the xylem under different weather conditions. We agree to your argument that RWU and SF have to be considered as interactive processes. Hence we expect the plants to adapt to climatic and site conditions. We expect that this adaptation is not a random process but some sort of optimisation. Based on the comments by Jesse Nippert on the issue of the concept of optimality, we have now avoided the optimality term in the document except for two specific citations. Here we rephrased the sentence to discern the plant's adaptation from the search for some "optimisation function", which describes this adaptation.

Page 16, line 12. Yes, wounding from sap flow sensors can indeed underestimate sap flux velocity, and nonhomogenous xylem depths can influence estimates of total transpiration rates, but it seems unlikely that these effects would be most noticeable during periods when both sap flux and RWU begin to decline. The authors allude to other factors in the previous paragraph (e.g. stem storage, leaf level transpiration) that offer more likely explanations for why correlations between RWU and sap flux correlations decrease as the soils dry out. Thank you for your evaluation of these influencing factors. We have revised the paragraph to now refer to the linear regression analysis and to propose the effect of wounding to be minor since the observed regression has a seasonal pattern but not a noticeable deviation from the regression. We removed the repeated mentioning from the discussion.

**Referee comment by Leander Anderegg**

In this manuscript, the authors pair soil moisture measurements that have high spatial and temporal resolution with tree sapflow measurements at two different sites to test whether such soil moisture measurements can be used to estimate daily transpiration and identify depths of root water uptake (RWU). They find promising similarities between sap flow and estimated RWU during a fairly wet period at their site with sandy soil, but worse correlations at a site with more heterogeneous soil characteristics and a time series that extended into a drier period. They also found interesting evidence for differences in the depth of RWU at the two study sites, though this is somewhat deemphasized in the text. While the estimated daily RWU uptake appears promising in some regards, they also found a confusing lack of relationship between RWU and soil matric potential calculated from soil moisture release curves and soil water content. All told, these results suggest that the method is promising but still has some kinks to be worked out, some of the largest probably relate to spatial heterogeneity at large scales (lateral variation over meters) and fine scales (inability to infer matric potential from soil moisture release curves on nearby soil samples.

This is an interesting manuscript that presents a promising approach to estimating transpiration and RWU at high temporal scale. However, my three main concerns are:

1) The writing and figures are extremely dense and sometimes confusing/contradictory. I had to read the Results at least two times, and often had to parse out individual sentences multiple times before I could begin to follow their meaning. Some of this could be due to a difference in fields (hydrology vs the plant ecophysiology terminology that I am more familiar with), but I would recommend a considerable expansion of the Results to explain the more complicated an nuanced findings and make this interpretable by a broad audience. I have given multiple suggestions below in the 'Specific Comments' section, but would general recommend a careful edit and clarification of the most complicated sentences in the Results. I also would recommend simplifying some of the figures by breaking out aspects into multiple panels rather than layering on 4-5 different sources of information that I found almost impossible to interpret simultaneously. In particular, Fig 6 and 12 are nearly impenetrable (and Fig 11 is also quite dense).

Thank you for your intense study of our manuscript and taking the challenge to dig out our messages so well. We gratefully received your suggestions and have now clarified and simplified the manuscript including some of the figures for better understanding.

2) The introduction oversells the novelty of monitoring soil moisture to estimate RWU dynamics and transpiration. True, the ability to monitor soil moisture with high enough precision to assess daily RWU is fairly novel and new, but people have been measuring soil moisture to estimate depths of RWU and understand transpiration budgets for decades! In fact, I would argue that gravimetric or volumentric soil moisture measurements are the original method for estimating transpiration (e.g. just to name a couple that come up with a quick google search: Denmead & Shaw (1962) "Availability of soil water to plants as affected by soil moisture content and meteorological conditions" Agronomy Journal; Novak (1987) "Estimation of soil-water extraction patterns by roots" Agricultural Water Management). Thus, I think it is important in the Introduction to stress that it is the precision of these measurements (allowing high temporal and spatial estimation of RWU) that is interesting, not the method and theory itself.

We generally agree to this point. We have revised the introduction and hope to be more concise about the goal of our study now: Although the connection of transpiration to soil moisture changes are well-known, we want to highlight the capability of our easily available technique for such analyses - given the level of precision and spatial coherence of the available soil moisture data.

3) I think the authors do a good job honestly discussing where their approach did not perform well, but I would both urge them to focus and structure the discussion around a coherent argument for what the key processes and attributes

are that screw up these measurements (e.g. what are the 4 biggest problems, list them out, and show us how you concluded that these are what is causing the method to fail at the Slate site and in dry soils).

Thank you for acknowledging our efforts. From our measurements we cannot distill a list of biggest problems. We revised the manuscript to make clear that we consider both RWU and SF as useful complementary measures to understand the transpiration process. As for our RWU algorithm we hope to now discuss better what its capabilities but also its limitations are.

I would also urge the authors to reconsider the framing and discussion around their 'Hypothesis 3'. It is currently framed as an open question whether tension gradients drive variation in root water uptake. And then Figure 11 is presented as evidence that this may not be the case. I think this is a misrepresentation of both where the field is at and what the confusing findings of Figure 11 represent. Plants can alter RWU via changes to root properties (changing aquaporin expression to alter root permeability) and root distribuitons, but they cannot physically fight potential gradients as the authors seem to suggest with Fig 11 and in the Discussion. Plants can ONLY extract and move water by moving it down a potential gradient, and there is no physical way the plant can be extracting and transpiring water from soils with a matric potential 10s-100s of MPa below 'permanent wilting point' (~1.5 MPa, or 4.2 (log10(hPa)). The general dogma (assuming +/- equivalent root resistances throughout the soil profile) that water uptake by roots should be proportional to the pressure difference and the root surface area/biomass should be used as a final test for the reliability of this method to estimate RWU, rather than using the data to test the dogma. In this case, I think it is painfully obvious that we have essentially no reliable way to convert water content to matric potential at the spatial and temporal scales that are relevant to these transpiration estimates. In fact, we're SO BAD at it, that it would appear that the Slate trees are extracting water from soil with a matric potential of « -10 MPa (when leaf water potential, the ultimate pressure differential driving water movement, is almost certinaly > -2 MPa). That tells me that there's a problem with the method, not the theory. However, recognizing this allows you to say something interesting about why we can't back calculate matric potential from these measurements (spatial heterogeneity in soil properties? Problems with rock fractions? Rock fractures that don't behave like soil samples used for dehydration curves?).

We completely dropped this aspect from the main part of the study because the methodology is problematic (as we explained in our initial replies), and it also does not contribute to the focus of our study. We left some of the details in Appendix B.

Specific comments: Pg 6 L11-14: Please explain a little more what you mean by 'NSE is a measure which is very sensitive to deviations from shape features" (perhaps you could add a day that does not pass this cuttof to Fig 4 to illustrate?), what cutoff of NSE you used, and how you arrived at that cutoff.

We rephrased the NSE description to make clear that we do not use the NSE as a cutoff for RWU calculations but rather as an additional evaluation how the identified soil moisture declines correspond to a mainly RWU-driven step shape. We added additional examples for such declines and the respective NSE values to Fig. 4.

Pg 7 L8-12 and pg10 L30 and Fig 7: I am very confused about what 'corrected' means. In the Methods, I interpreted 'Corrected' to mean RWU extrapolated from the linear regression through the nightly data (magenta line in Fig 4a). But in Fig 7 the 'not corrected' values (blue points) are higher than the 'corrected' values (colored points), which tells me I'm getting confused somewhere. Please clarify this in Fig 4, and Fig 7 and the associated Results text (pg 10 L30).

Thank you for pointing this out. We took care to clarify this in the revision. We now call the approach without the regression through the nightly data the "simplified" approach and then introduce our extended approach with the regression step by step in the methods section 3.1. As we continue the analyses with the extended approach, we now also mention explicitly when we do the comparison of both approaches (section 4.4).

*Pg 7 L20-29: This paragraph about turning sap velocity into sap flux is very confusing. I did not understand it until I scrutinized Fig 5. Please rewrite/clarify. Also, in the Fig 5 legend/caption it is worth noting that the "5mm, 18mm, and 30mm" are depths from the outside of the tree (or inside of inner bark? Not sure which).*

We have completely revised this section (3.3) to guide the reader though the calculations step by step, alongside Fig. 5. We hope the procedure can be understood more easily now.

Figure 6: I had a very difficult time extracting the desired inferences from this figure. The shading (which varies per site, over time, and in different soil layers) is almost impossible to see and interpret (not to mention some of the colors become colors used for other soil depths when shaded) yet are referenced multiple times in the text. Also, the stacked bar plots make it almost impossible for me to interpret which depths are providing RWU, mostly I just take away total bar height. I would recommend 1) breaking out the information about how well the RWU estimation likely worked into another panel or method other than shading (filled versus unfilled bars/symbols, perhaps?). and 2) either finding a more holistic way of showing depth information (e.g. coloring whole bars by the weighted average depth of RWU) or just making a different panel that showed line graphs of uptake by depth through time. In fact, I would potentially advocate for breaking out the depth of extraction information into a new figure altogether.

We have revised the figure and removed the shading. Using a line plot did not turn out to be helpful. Instead we have also clarified the processing of the data to derive the plot more clearly in the new Fig. 3. We hope that these revisions have led to a much easier comprehension of our approach and data. As a summarising information about the soil water availability, we added this data to the figure.

*Pg 12 L14-15: I don't quite understand what data are being compared in this sentence "Comparing RWU correlation between the two sites, applying the nocturnal correction improves Spearman rho form 0.42 to 0.52. KGE remains almost the same with 0.27 increasing to 0.3." All data from both sites (if so, why is this a useful comparison)? Or somehow site-level averages?*

We clarified the difference between our approach including the nightly recharge and the simplified approach of simply assessing the soil moisture reduction between two days in the methods section. We moved the topic to a separate discussion section (4.4) to clarify what we compare and conclude. Basically, we repeat the correlation calculations between RWU and SF and between the two sites with the simplified approach and compare the resulting measures.

Section 4.3 – I think this section is very cool, but I understood very little of the text. What does "a diffuse redistribution into the surrounding soil aggregates" mean and why can it be "seen as parallel declines. . .in the different depth layers"? Please explain more what "flashy transport through the macroporous soils and fill-and-spill mechanisms of subsurface pools" means, and much more importantly how this analysis influences our interpretation of the method for assessing RWU in this site. Clearly you learned something interesting and highly relevant (possibly that helps us interpret Fig 11?), but I do not understand what it is based on the current text. For instance, I have no idea what these sentences mean and how they relate to Fig 10b "Here, roots are likely to grow along joints and fractures, where eventwater can be stored with little effect on the bulk soil moisture. As such, the measurements might miss parts of the active rhizosphere."

We moved this part to the site description (section 2.2) as it gives some background to the contrasting pedological conditions of the sites, which are likely to affect water availability to the tree. We have revised the paragraph and hope it is easier to understand now. However, some of the jargon persisted as standing terminology.

Section 4.4 – See above comments about interpretation and framing of these results. Also, the current Figure 11 is nearly impossible for me to interpret. I would recommend displaying SOME aspect of this information in multiple panels. (e.g. maybe splitting the soil columns up into three depths and displaying them as separate panels so you can color by SF). Also, the units/label on the x axis of this figure is confusing to me. And honestly, after reading the text of this section 4 times, I still don't have any idea what it means. I can't even decipher it enough to make suggestions on how to clarify it. I don't know what the referenced 'reactions' are and how I'm supposed to assess them in the figure. Moreover, I do not at all see the 'correlation of matric potential and depth' that supposedly exists in slate site.

We will re-evaluate Fig. 11. As you point out, we also need to rework the whole argument showing that it is not the plants sourcing at high flux rates against physiologically impossible tensions but the conversion of soil moisture into matric potentials, which does not represent the state around the roots. We will take care of this in the revision.

We have moved this to the appendix and rephrased most of it.

Pg 14 L14: This sentence "At the same time, we pointed out considerable limitations to the approach with respect to soil water state (no detectable signal during low moisture periods) and soil properties (high variability in heterogeneous soil profiles)" is the most interesting sentence of the discussion to me, but comes out of no where and needs much more explanation. In order for me to follow your train of thought, I require much more explanation...

Thank you for highlighting this lack of reference. We have revised Fig. 6 to include plant available soil water and an evaluation of the detected steps with NSE>0.5. Going into more detail about the determination of our approach reveals that it cannot be explained with soil water availability alone but that the seasonal state of the tree has also a strong effect. We have revised the presentation of the results and the discussion accordingly.

Figure 12 and associated text of Section 5.1: I had an extremely hard time interpreting this figure. Please 1) remove the red bars for total extraction to new panels (two axes y- axes with different interpretations is much more than my brain can handle). 2) Explain what the NSC cutoffs indicate, and what the larger blue bar for 'all detected' is and why the inset bars for different detection thresholds do not sum to it 3) Put panel A and B on the same axis (e.g. 0%-90%) and switch the big numbers to be % of days and little numbers to be # of days. Also, how does Fig 12 show "The RWU derivation function appears to perform very well in general and can be used to evaluate a broad range of diurnal changes in soil moisture (Fig. 12)." (L1-2). Moreover, this sentence doesn't really make sense to me "Unlike the first impression in Fig. 6, the proportion of steps with higher uncertainty about the actual fit of the shape with the assumptions is higher in the slate site data, which is in line with the lower overall RWU detection there." Could you explain what you mean by "higher uncertainty about the actual fit"? Also, how "uncertainty" and rate of "overall detection" differ? Throughout this section, please be much more explicit about the site, times, and layers you are referring to when, for example, you write "Under somewhat ideal conditions with soil moisture sensors and roots in good contact with a rather homogeneous soil matrix and sufficient soil water availability, the diurnal steps are identified and evaluated with great confidence." Finally, this feels like it should be in the Results, perhaps even near Figure 1, rather than in the Discussion.Pg 15- L5: I think it's worth explicitly mentioning the take-away from Figure C1: that flux amount is unrelated to how well the step function fits the daily soil moisture pattern.

Thank you for pointing out the difficulty of this figure. We have opted to leave the figure as it is because we considered it important to have the red bars directly in the figure to correctly interpret the significance (or lack thereof) of the blue bars. However, we have greatly extended the figure caption, guiding the reader through the figure in hopefully enough steps now. We also added more explanation to the respective text paragraph. With respect to the proposed equal scaling of the y-axes to the respective proportions, we have left the figure with reference to separate counts. By doing so we hope to avoid misleading interpretations of the comparison of the different observation periods. The sand site does not include the later phase of the season.

*Pg* 16-L25-35: See my comments about Hypothesis 3 and Figure 11. Also, the sentences at L28 ("At the sandy site...") seem confusing and almost self contradictory to me.

We revised hypothesis 3 to simply highlight the site influence on RWU and SF (as a research question) without any further assumptions. Consequently, we removed the whole consideration of matric potential from the main manuscript to the appendix following your suggestions. We refer to some of it in the discussion but point to the uncertainty in these calculations and refrain from any interpretation.

**References**

[revised manuscript text omitted]

---

## Author Response (AR2)

Reply to Associate Editor Decision of MS bg-2019-466:
Estimates of tree root water uptake from soil moisture profile dynamics by Conrad Jackisch et al.

Dear Chris,
Dear Copernicus Team.

Thank you again for handling our manuscript and recruiting such excellent referees.
We have addressed the technical corrections as follows:

*Figure 1B – This might be more clear as ''Comparison of several (scaled) soil moisture declines demonstrating a range of Nash-Sutcliffe-Efficiencey (NSE) scores compared to an artificial (''ideal'') reference step.''*

This appears to refer to Fig. 4B. We have changed the caption as proposed.

*Pg7 L11 – the assumption that hydraulic redistribution processes 'remain active during the day' strikes me as an unrealistic assumption if the fluxes are plant mediated. If there is some purely physical water movement (capillary flow from a saturated layer, downward flow from a higher soil layer) this might be true, but plant-mediated 'hydraulic redistribution' decreases markedly as soon as the plant water potentials begin to fall in the morning (because the water potential gradient drives water up the plant rather than out of the roots).*

We agree to the reviewer and have specified the redistribution fluxes as ''physical'' (now P8L5). Moreover we added a paragraph to Sec. 5.1 as follows:
With respect to our assumption of nocturnal hydraulic soil water redistribution and its continuation over the day, there remains room for further research and refinement. Especially when hydraulic redistribution is plant-mediated and given the multiple occasions with slightly negative nocturnal soil moisture slopes, our assumption might not hold.

*Pg13 L1 – Perhaps it's worth including one sentence of interpretation here rather than just turning readers loose on the Appendix. Something like "Although higher RWU fluxes were generally associated with more step-like soil moisture shapes and higher NSEs, there were still a number of high RWU days at both sites with poor NSEs, indicating additional complicating factors beyond higher detection with higher fluxes (Appendix Fig. B1).''*

Thank you for the suggestion. We have included the interpretation sentence here (P13L1).

*Fig 8 & Pg16 L6-12– it looks like most of the 'simplified RWU' calculations are actually larger than the hydraulic redistribution-corrected fluxes. Does this mean that most nightly trends were actually negative (rather than positive as shown in Fig 2)? If this is the case, it seems like it is not actually hydraulic redistribution that is at play, but rather refilling of plant capacitance or some such subtle overnight water withdrawal.''*

The reviewer definitely has a point here. This further underlines a critical revision of our assumption as pointed out before. We have added a respective paragraph at the end of Sec. 4.4:
However, the slightly positive bias of the simplified approach (light blue regressions in Fig. 8 above red regression line) points to cases with negative nocturnal changes in soil moisture. This could also be explained as refilling process of the plant's capacitance.

We came across another minor mistake in our calculations: RWU was not reported in mm but in Δvol.%. To convert the values we have to multiply Δvol.% with the soil moisture measurement increment of 200 mm. This results in factor 2 values, obviously. Since we did not refer to absolute values until the linear regression (Fig. 8) and since the applied linear regression is not affected by a scaling factor, this error does not have any implications on the outcomes of our study. We have updated all graphs which report RWU in mm/day and we have updated the reduced mean rhizosphere radius in Sec. 4.3. Moreover, we have modified the rhizosphere radius reference statement to existing studies in Sec. 5.2 (P18L13ff.). We have carefully checked all results and provide a Jupyter Notebook along the paper for transparency and own calculations.

Thank you and all the best,
on behalf of all co-authors,
Conrad and Sibylle